# GLOBALLY 1-LIPSCHITZ ATTENTION WITHOUT SEQUENCE-LENGTH DEPENDENCE

## ABSTRACT

Self-attention powers modern deep learning; however, dot-product attention is not globally Lipschitz, which limits stability and robustness. Prior fixes enforce Lipschitz continuity by changing the geometry of attention, which departs from the standard mechanism and still yields bounds that scale poorly with sequence length or spectral norms. We introduce AttLip, a new attention block that derives coefficients from a convex potential and realizes them through an implicit proximal update, guaranteeing architectural stability. This design ensures the entire block is firmly non-expansive and thus *unconditionally 1-Lipschitz*, independent of sequence length or parameter norms, while remaining structurally close to dot-product attention. Building on monotone operator theory, we establish its contractive properties and develop efficient first-order solvers. Experiments on OpenWebText show that AttLip achieves meaningful token mixing and retains learning capacity.

## 1 INTRODUCTION

Self-attention (Bahdanau et al., 2015; Vaswani et al., 2017) is the driving force behind modern deep learning, powering large language models (LLMs) (Brown et al., 2020; Bubeck et al., 2023) and vision transformers (ViTs) (Dosovitskiy et al., 2021; Radford et al., 2021). Despite its success, its mathematical behavior is less well understood than that of classical layers such as convolutions, for which operator norms and Lipschitz constants are well characterized (Sedghi et al., 2019; Delattre et al., 2023). In particular, dot-product attention is not globally Lipschitz (Kim et al., 2021), meaning small input perturbations can induce disproportionately large output changes. This raises concerns for optimization stability (Qi et al., 2023) and robustness (Zhou et al., 2022; Cissé et al., 2017). The fragility of neural networks to imperceptible adversarial perturbations (Biggio et al., 2013; Szegedy et al., 2014) has motivated certified defenses that provide provable robustness guarantees (Raghunathan et al., 2018; Wong & Kolter, 2018). A principled route to certification is to enforce Lipschitz continuity: if the global Lipschitz constant of a network is bounded, one can derive certified robustness radii around any input. Two strategies have emerged. Randomized smoothing (Lécuyer et al., 2019; Cohen et al., 2019) scales to large datasets but is constrained by impossibility results (Yang et al., 2020). Alternatively, networks can be built from 1-Lipschitz layers, ensuring tight global control (Cissé et al., 2017; Li et al., 2019; Anil et al., 2019; Trockman & Kolter, 2021b; Singla & Feizi, 2021). While well studied for convolutions and linear layers, Lipschitz control of attention remains largely unresolved. Recent work has addressed this gap along two directions. First, Lipschitz-constrained variants such as $\ell_2$-attention (Kim et al., 2021; Dasoulas et al., 2021) enforce global continuity but yield bounds that scale poorly with sequence length. Second, local smoothness analyses of dot-product attention (Castin et al., 2024; Xixu, 2023) provide finer control but again degrade unfavorably with sequence length. As a result, standard attention remains highly expressive but inherently non-Lipschitz, while Lipschitz alternatives offer guarantees that become vacuous at realistic context sizes.

We propose to bridge this gap with an implicit attention block, denoted AttLip. The key idea is to reinterpret attention coefficients as the gradient of a convex potential (Meunier et al., 2022) and to realize them through an implicit proximal update. This construction guarantees that the block is firmly non-expansive and thus globally 1-Lipschitz, independent of sequence length or parameter norms. Intuitively, the convex potential encodes pairwise interactions, while the proximal

formulation enforces stability by design. In this way, AttLip provides rigorous Lipschitz control without abandoning the expressive token-mixing behavior that makes attention effective. Our main contributions are:

1. **First global 1-Lipschitz attention block independent of sequence length.** We introduce AttLip, an implicit attention mechanism derived from convex potential theory and realized through a proximal operator, ensuring strict 1-Lipschitzness by construction.

2. **Theoretical guarantees and comparison.** We develop an operator-theoretic analysis of AttLip, establish its contractive properties, and compare its guarantees to those of $\ell_2$-attention (Kim et al., 2021), highlighting the independence from sequence length.

3. **Practical trainability and empirical validation.** We show that AttLip can be trained efficiently with first-order solvers and retains meaningful learning capacity. Experiments on Transformer pretraining with OpenWebText validate its effectiveness in practice.

## 2 RELATED WORK

**Global Lipschitz control in deep networks.** A function $f$ is $L$-Lipschitz w.r.t. the $\ell_2$ norm if $\|f(x) - f(x + \delta)\|_2 \leq L\|\delta\|_2$, $\forall x, \delta$, meaning that its output cannot change faster than a constant multiple of the input perturbation. The smallest such constant is the global Lipschitz constant $L_2(f)$. Controlling $L_2(f)$ has been shown to improve generalization, through spectrally-normalized margin bounds (Bartlett et al., 2017), and is also central to certified robustness: the certified radius of a classifier scales proportionally to its margin and inversely with $L_2(f)$ (Tsuzuku et al., 2018). Because computing $L_2$ exactly is intractable for deep models, two families of approaches have emerged. *Estimation/certification:* AutoLip and SeqLip yield generic and structured upper bounds, while convex relaxations based on quadratic constraints provide tight certificates at scale (Virmaux & Scaman, 2018; Fazlyab et al., 2019). Operator-theoretic analyses further give a posteriori layerwise certificates by modeling activations as averaged (firmly non-expansive) operators (Combettes & Pesquet, 2020). *Design by construction:* spectral normalization constrains operator norms during training (Miyato et al., 2018; Farnia et al., 2019); Parseval regularization encourages near-orthonormal weights (Cissé et al., 2017); exact singular-value characterizations enable norm control for convolutions (Sedghi et al., 2019); and orthogonal/near-isometric parameterizations (e.g., Cayley) preserve norms in deep convnets (Trockman & Kolter, 2021a; Prach & Lampert, 2022; Boissin et al., 2025). On the function-class side, GroupSort networks with norm-constrained weights are universal Lipschitz approximators with refined approximation guarantees (Anil et al., 2019; Tanielian et al., 2021). Dynamical-systems and control viewpoints motivate *implicit/contractive* layers; in particular, convex potential layers realize transformations as resolvents/proximal maps, making them 1-Lipschitz by design (Meunier et al., 2022; Araujo et al., 2023; Wang & Manchester, 2023). Monotone operator theory underpins these guarantees via firm non-expansiveness of resolvents (Bauschke & Combettes, 2017).

**Lipschitzness of self-attention.** Standard dot-product self-attention is not globally Lipschitz: unbounded logits can induce sharp changes in softmax weights. Two main directions address this limitation. (i) *Local smoothness analyses* bound the *local* Lipschitz constant (or sensitivity), typically with dependence on input norm and sequence length (e.g., worst-case local constants scaling like $\widetilde{\mathcal{O}}(\sqrt{n})$), which limits certification at realistic context sizes (Castin et al., 2024; Havens et al., 2024a). (ii) *Lipschitz-by-modification* enforces global control by altering the mechanism, most commonly by replacing dot products with negative squared distances ($\ell_2$-*attention*) or by score normalization (Kim et al., 2021; Dasoulas et al., 2021). While theoretically sound, these variants change attention geometry/distributional behavior, and their bounds often grow with sequence length. Beyond analysis, certified robustness for Transformer-style models has been advanced by (post-hoc) proximal/projection schemes for ViTs that tighten bounds and can swap dot-product attention for $\ell_2$-attention without full retraining (Gupta & Verma, 2023); by contractive-flow formulations integrating attention (Mukherjee et al., 2023); and by 1-Lipschitz Transformer variants under additional constraints (Xu et al., 2023). Vision-focused studies further document robustness/sensitivity patterns (Zhou et al., 2022; Qi et al., 2023). Overall, existing results either provide *local*, length-dependent guarantees for dot-product attention, or achieve *global* control at the cost of changing the attention geometry and/or introducing length-dependent constants.

**Adversarial robustness in NLP.** Modern NLP systems built on self-attention are highly capable yet *fragile* to small input edits. A large body of work shows that character flips, synonym substitutions, short appended phrases, or crafted prompt suffixes can reliably destabilize model behavior, motivating *stability by design* through architectural control of input–output sensitivity.

Early attacks established that tiny, often imperceptible edits can flip predictions. Ebrahimi et al. (2018) introduced *HotFlip*, using gradients to propose character- and word-level substitutions that efficiently degrade accuracy. In a black-box setting, Jin et al. (2020) showed that meaning-preserving synonym swaps reliably fool strong classifiers, including BERT. Follow-ups refined search constraints (e.g., part-of-speech consistency, semantic similarity thresholds) and explored gradient-free strategies and genetic algorithms for discrete sequences (Alzantot et al., 2018; Liang et al., 2018). Tooling such as Morris et al. (2020) (TEXTATTACK) standardized constraints, transformations, and search procedures, enabling reproducible comparisons across models and tasks.

Beyond instance-specific edits, Wallace et al. (2019) discovered *universal adversarial triggers*: short token sequences, optimized once, that when appended consistently elicit targeted behaviors across tasks and architectures. This foreshadowed *jailbreak* suffixes for instruction-tuned LLMs: Zou et al. (2023) ("GCG") demonstrated automatically optimized suffixes that transfer across open and proprietary models, reliably bypassing guardrails.

On the defense side, adversarial training and robust fine-tuning improve empirical robustness but lack worst-case guarantees (Jiang et al., 2020; Aghajanyan et al., 2021). Randomized smoothing provides probabilistic certificates at scale but faces intrinsic limits on achievable radii and sensitivity to distributional mismatch (Lécuyer et al., 2019; Cohen et al., 2019; Yang et al., 2020). A parallel line of work develops *certified defenses* tailored to word substitutions, including interval-bound and smoothing methods (Jia et al., 2019; Ye et al., 2020; Zhao et al., 2022), which demonstrate non-trivial certified robust accuracy on IMDB and YELP benchmarks.

These threads highlight the need for *architectural* stability mechanisms, specifically block-level non-expansiveness that delivers global (sequence–length–independent) guarantees.

## 3 BACKGROUND ON SELF-ATTENTION AND CONVEX POTENTIAL LAYERS

We focus on *self*-attention with *global* context (not cross- or causal attention), which is the regime compatible with our convex/PSD construction.

**Standard attention.** Consider an input matrix $\boldsymbol{X} \in \mathbb{R}^{N \times d}$ with $N$ tokens of dimension $d$. For each head $h$, define the score matrix

$$\boldsymbol{S}_{ij}^{\text{std},h} = \frac{(\boldsymbol{x}_i \boldsymbol{W}_Q^h)(\boldsymbol{x}_j \boldsymbol{W}_K^h)^\top}{\sqrt{d_h}}, \qquad i,j = 1, \ldots, N. \tag{1}$$

Standard multi-head dot-product self-attention (Vaswani et al., 2017) is then

$$\text{Att}(\boldsymbol{X}) = \text{concat}_{h=1}^H \left( \text{softmax}_{\text{row}} \big( \boldsymbol{S}^{\text{std},h} \big) \boldsymbol{X} \boldsymbol{W}_V^h \right) \boldsymbol{W}^O, \tag{2}$$

where $\text{softmax}_{\text{row}}$ applies the softmax function independently to each row, $\boldsymbol{W}_Q^h, \boldsymbol{W}_K^h, \boldsymbol{W}_V^h \in \mathbb{R}^{d \times d_h}$ are the query, key, and value matrices for head $h$, $\boldsymbol{W}^O \in \mathbb{R}^{H d_h \times d}$ is the output projection, and $\text{concat}$ denotes concatenation along the feature dimension.

While extremely effective in practice, this operator is not globally Lipschitz in the $\ell_2$ norm. The attention weights depend exponentially on pairwise dot products, which scale quadratically with $\|\boldsymbol{X}\|_2$. Small perturbations of $\boldsymbol{X}$ near score ties can cause abrupt changes in the attention distribution, and this effect grows without bound as $\|\boldsymbol{X}\|$ increases. However, when input $\boldsymbol{X}$ is bounded, one can derive a local Lipschitzness property for self-attention.

**Theorem 1** (Multi-head adaptation of the local Lipschitz bound; adapted from Xixu (2023))**.** *Consider a multi-head self-attention layer with $H$ heads concatenated and a final output projection $\boldsymbol{W}_O$. For $\boldsymbol{X}$ in the Frobenius ball $B_2(\boldsymbol{X}_0, \delta_0)$, define for each head*

$$L_h = N(N+1)(\|\boldsymbol{X}_0\|_F + \delta_0)^2 \Big[ \|\boldsymbol{W}_V^h\|_2 \|\boldsymbol{W}_Q^h\|_2 \|(\boldsymbol{W}_K^h)^\top\|_2 + \|\boldsymbol{W}_V^h\|_2 \Big].$$

*Then the multi-head attention mapping* Att *is locally Lipschitz on* $B_2(\boldsymbol{X}_0, \delta_0)$ *with*

$$\mathrm{L}_2\big(\mathrm{Att}, \boldsymbol{X}_0\big) \ \leq \ \|\boldsymbol{W}_O\|_2 \left(\sum_{h=1}^{H} L_h^2\right)^{1/2}. \tag{3}$$

In parallel, Castin et al. (2024) also derive worst-case bounds on the local Lipschitz constant of self-attention, Havens et al. (2024b) develop a fine-grained local sensitivity analysis of dot-product self-attention, showing that while the layer is not globally Lipschitz, its local sensitivity to small $\ell_2$ perturbations can be bounded and used to obtain certified robustness guarantees on vision transformers. However, these guarantees are only valid for small perturbation budgets. However, local Lipschitz analyses of standard dot-product attention remain limited, as the constants depend on the input norm.

**Modified Lipschitz attention.** This phenomenon was formalized by Kim et al. (2021), who proposed an alternative $\ell_2$-attention mechanism that is provably Lipschitz continuous. Specifically, they replace dot-product similarity with negative squared $\ell_2$ distances between transformed tokens.

For each head $h$, define the $\ell_2$-attention score matrix

$$\boldsymbol{S}_{ij}^{\mathrm{Kim},h} \ = \ -\frac{1}{\sqrt{d_h}} \|\boldsymbol{x}_i \boldsymbol{W}_Q^h - \boldsymbol{x}_j \boldsymbol{W}_K^h\|_2^2, \qquad i, j = 1, \dots, N, \tag{4}$$

and the multi-head $\ell_2$-attention layer

$$\mathrm{Att}_{\ell_2}(\boldsymbol{X}) \ = \ \mathrm{concat}_{h=1}^{H}\left(\mathrm{softmax}_{\mathrm{row}}\big(\boldsymbol{S}^{\mathrm{Kim},h}\big)\,\boldsymbol{X}\boldsymbol{W}_V^h\right)\boldsymbol{W}^O. \tag{5}$$

Note that for this layer to be Lipschitz continuous, weight tying is necessary: $\boldsymbol{W}_Q^h = \boldsymbol{W}_K^h$.

**Theorem 2** (Lipschitz bound for $\ell_2$-attention (Kim et al., 2021))**.** *The operator* $\mathrm{Att}_{\ell_2}$ *is Lipschitz continuous in the* $\ell_2$ *norm, with Lipschitz constant bounded as*

$$\mathrm{L}_2(\mathrm{Att}_{\ell_2}) \ \leq \ \frac{\sqrt{N}}{\sqrt{d_h}}\Big(4\,\Phi^{-1}(N-1) + 1\Big)\left(\sqrt{\sum_{h=1}^{H} \|\boldsymbol{W}_Q^h\|_2^2 \|\boldsymbol{W}_V^h\|_2^2}\right)\|\boldsymbol{W}^O\|_2, \tag{6}$$

*where* $\Phi(x) = x\exp(x+1), \quad x > 0$, *and* $\Phi^{-1}(N-1) = W_0\big(\frac{N-1}{e}\big)$, *where* $W_0$ *is the principal branch of the Lambert W function.*

While theoretically sound, the bound scales in a nontrivial way with $N$, and the operator departs from dot-product attention both geometrically (distance-based vs. inner-product) and in its distributional behaviour. Even in this Lipschitz formulation, the bound remains intricate and explicitly tied to the sequence length $N$, which is problematic in practice, especially when contrasted with simpler constructions where the Lipschitz constant can be exactly 1 (Trockman & Kolter, 2021a; Meunier et al., 2022; Miyato et al., 2018).

**Scalability of Lipschitz bounds.** The analytic bound for $\ell_2$-attention proposed by Kim et al. (2021) grows with the sequence length $N$ as $\mathcal{O}\left(\frac{\sqrt{N}}{\sqrt{d_h}}\big(4\phi^{-1}(N-1)+1\big)\right) \ \sim \ \mathcal{O}(N\log N)$, whereas local Lipschitz constants obtained from the Jacobian typically exhibit quadratic growth, $\mathcal{O}(N^2)$. This scaling becomes problematic given the sequence lengths used in practice: BERT is trained with up to 512 tokens (Devlin et al., 2019), GPT-2 with 1024 (Radford et al., 2019), GPT-3 with 2048 (Brown et al., 2020), and the LLaMA family ranges from 2048 in LLaMA-1 to 4096 in LLaMA-2 and 16k in LLaMA-3 (Touvron et al., 2023). More recent proprietary models push these limits much further, with GPT-4 and Claude 2 supporting context windows of 32k tokens, and specialized versions extending to over 100k. Although the Kim et al. (2021) bound is formally subquadratic, inserting these practical sequence lengths produces Lipschitz constants of astronomical scale, far beyond any interpretable range. These results highlight that such bounds should be viewed primarily as formal guarantees rather than tight measures of effective sensitivity.

This reveals a fundamental tension: standard attention is powerful but non-Lipschitz, while Lipschitz alternatives exist but induce different behaviour. Our objective is therefore to design an attention-like operator that is provably Lipschitz yet retains token-mixing behavior.

## 3.1 Contractivity from convex potential flow

The input matrix $\boldsymbol{X} \in \mathbb{R}^{N \times d}$ is written in vectorized form as $\boldsymbol{x} = [\boldsymbol{x}_1^\top \dots \boldsymbol{x}_N^\top]^\top \in \mathbb{R}^{Nd}$ with $\boldsymbol{x}_i \in \mathbb{R}^d$. Convex gradient flows are known to generate contractive dynamics in Hilbert spaces, which provides a dimension-free stability guarantee.

**Theorem 3** (Contractivity of convex gradient flows (Brézis, 1973))**.** *Let $f : \mathcal{H} \to (-\infty, +\infty]$ be proper, lower semicontinuous, and convex on a Hilbert space $\mathcal{H}$. Consider the (sub)gradient flow*

$$\frac{d\boldsymbol{x}(t)}{dt} + \partial f(\boldsymbol{x}(t)) \ni 0.$$

*Then the associated semigroup is nonexpansive:*

$$\|\boldsymbol{x}(t) - \boldsymbol{y}(t)\| \le \|\boldsymbol{x}(0) - \boldsymbol{y}(0)\| \qquad \forall\, t \ge 0,$$

*for any two solutions $\boldsymbol{x}(\cdot), \boldsymbol{y}(\cdot)$. In particular, if $f$ is differentiable, the gradient flow $\frac{d\boldsymbol{x}(t)}{dt} = -\nabla f(\boldsymbol{x}(t))$ is 1-contractive.*

This theorem formalizes the fact that the gradient of a convex potential defines a contractive dynamical system. A more general operator-theoretic result due to Rockafellar (Rockafellar, 1976) shows that the resolvent of any maximal monotone operator is firmly nonexpansive, and therefore 1-Lipschitz. Together, these results provide both the continuous-time and discrete-time foundations for constructing contractive mappings from convex potentials.

**Application to neural networks.**   Building on these principles, Meunier et al. (2022) proposed convex-potential layers where the vector field is taken as $F = -\nabla f$ with $f$ convex. By Theorem 3, the resulting continuous-time flow is contractive, and discretization yields a 1-Lipschitz residual mapping. Their implementation uses the *explicit Euler scheme*,

$$\boldsymbol{x}_{t+1} = \boldsymbol{x}_t + \eta F(\boldsymbol{x}_t),$$

which ensures non-expansiveness under smoothness assumptions on $f$. This line of work is conceptually related to *Input Convex Neural Networks* (ICNNs) (Amos & Kolter, 2017), which enforce convexity through architectural constraints, though here convexity is leveraged to design contractive flows.

# 4 Designing Lipschitz Attention Mechanism

We now present our construction of a globally 1-Lipschitz attention block. The key idea is to reinterpret attention scores as the gradient of a convex potential, and to realize the update through an *implicit proximal step*. This ensures contractivity by design, independent of sequence length or parameter norms.

## 4.1 From residual updates to convex potentials

**Residual formulation.**   In the *post-norm* Transformer (Vaswani et al., 2017), ignoring LayerNorm for clarity, the self-attention block reduces to a residual update

$$\boldsymbol{X}_{t+1} = \boldsymbol{X}_t + \text{Att}(\boldsymbol{X}_t)\boldsymbol{W}_O,$$

with output projection $\boldsymbol{W}_O \in \mathbb{R}^{d_h \times d}$. This is exactly an explicit Euler step of the ODE

$$\frac{d\boldsymbol{X}(t)}{dt} = F(\boldsymbol{X}(t)), \qquad F(\boldsymbol{X}) = \text{Att}(\boldsymbol{X})\,\boldsymbol{W}_O,$$

with step size $\eta = 1$. Similar dynamical-systems interpretations have been explored in neural ODEs, equilibrium models, and convex potential layers (Chen et al., 2018; Bai et al., 2019; Meunier et al., 2022).

**Convex potential formulation.** We now instantiate this principle for self-attention by defining a convex potential whose gradient has an attention-like structure. Using Theorem 3 we have:

**Proposition 1** (Convex Potential for Attention Flow). *Define the convex attention potential*

$$f(\boldsymbol{x}) \;=\; \tfrac{1}{2} \sum_{i=1}^{N} \mathrm{LogSumExp}\big(S_{i,:}^{convex}(\boldsymbol{x})\big),$$

*where the score matrix is given by* $S_{ij}^{convex}(\boldsymbol{x}) = (\boldsymbol{x}_i + \boldsymbol{x}_j)^{\top} \boldsymbol{W}^{\top} \boldsymbol{W} (\boldsymbol{x}_i + \boldsymbol{x}_j)$.

*Letting* $\alpha_{i,j}(\boldsymbol{x}) = \mathrm{softmax}_j\big(S_{i,:}^{convex}(\boldsymbol{x})\big)$ *denote the row-wise softmax weights, the gradient of* $f$ *takes the attention-like form*

$$\nabla f(\boldsymbol{x})_k = \Big(1 + \sum_{i=1}^{N} \alpha_{i,k}(\boldsymbol{x})\Big) \boldsymbol{W}^{\top} \boldsymbol{W} \boldsymbol{x}_k \;+\; \sum_{i=1}^{N} \big(\alpha_{i,k}(\boldsymbol{x}) + \alpha_{k,i}(\boldsymbol{x})\big) \boldsymbol{W}^{\top} \boldsymbol{W} \boldsymbol{x}_i.$$

*Therefore, the induced gradient flow* $\frac{d\boldsymbol{x}}{dt} = -\nabla f(\boldsymbol{x})$ *is contractive.*

Our construction enforces $\boldsymbol{W}_Q = \boldsymbol{W}_K = \boldsymbol{W}$ as in Kim et al. (2021) and $\boldsymbol{W}_V = \boldsymbol{W}^{\top} \boldsymbol{W}$, yielding a symmetric score and attention matrix. This parameter sharing departs from standard attention but is what guarantees convexity. See Algorithm 4 for efficient way to compute $\nabla_f(\boldsymbol{x})$.

**Choosing a discretization scheme.** In our setting, the convex potentials of interest involve quadratic forms and log-sum-exp terms, which are convex but generally non-smooth. For such functions, explicit Euler no longer guarantees stability. We therefore adopt the *implicit Euler scheme*,

$$\boldsymbol{x}_{t+1} = \boldsymbol{x}_t - \eta \nabla f(\boldsymbol{x}_{t+1}), \tag{IE}$$

which remains contractive under convexity alone and can be interpreted as an implicit layer (Bai et al., 2019). Although this step requires solving a nonlinear system, we reinterpret it as a proximal operator, preserving non-expansiveness while allowing efficient first-order solvers in practice.

## 4.2 CASTING AS A PROXIMAL PROBLEM

The implicit Euler step (Eq. IE) involves the unknown $\boldsymbol{x}_{t+1}$ on both sides. It is equivalent to applying the *proximal operator* of $f$:

$$\boldsymbol{x}_{t+1} \;=\; \underset{\eta f}{\mathrm{prox}}(\boldsymbol{x}_t) \;=\; \underset{\boldsymbol{x} \in \mathbb{R}^{Nd}}{\arg\min} \Big\{ f(\boldsymbol{x}) \;+\; \tfrac{1}{2\eta} \|\boldsymbol{x} - \boldsymbol{x}_t\|_2^2 \Big\}. \tag{$P_t$}$$

This convex reformulation ensures that the update is well posed, admits a unique minimizer, and by maximal monotone operator theory the map $\mathrm{prox}_{\eta f} = (I + \eta \nabla f)^{-1}$ is firmly non-expansive and thus 1-Lipschitz (Bauschke & Combettes, 2017). Hence the implicit attention layer is contractive by design. This reformulation shows that the implicit step is a proximal operator of $f$, which not only guarantees well-posedness and contractivity, but also makes the update computationally tractable: instead of inverting a large Jacobian, one can solve the convex subproblem ($P_t$) efficiently with first-order methods.

**Non-smooth convex optimization.** The potential $f$ is convex but not globally smooth. Nevertheless, on any bounded sublevel set, its gradient is Lipschitz, which enables local convergence guarantees. We therefore minimize $\phi_t(\boldsymbol{x}) := f(\boldsymbol{x}) + \frac{1}{2\eta}\|\boldsymbol{x} - \boldsymbol{x}_t\|_2^2$ by gradient descent with updates

$$y^{(t+1)} \leftarrow y^{(t)} - \lambda_t \big(\nabla f(y^{(t)}) + \tfrac{1}{\eta}(y^{(t)} - \boldsymbol{x}_t)\big).$$

When a local Lipschitz constant $L_2(f, R)$ on the relevant sublevel set is available, any fixed step $\lambda \in \big(0, 2/(L_2(f, R) + 1/\eta)\big)$ ensures linear convergence. Otherwise, Armijo backtracking automatically selects a valid $\lambda_t$ at each iteration. In both cases, the iterates converge linearly to $\mathrm{prox}_{\eta f}(\boldsymbol{x}_t)$. The following is a direct application of standard results from convex optimization (e.g., Nesterov (2018), Bauschke & Combettes (2017), Bertsekas (1999)). We restate this standard result for completeness: it ensures the proximal subproblem can be solved to arbitrary precision, so the 1-Lipschitz property of $\mathrm{prox}_{\eta f}$ carries over to the implemented layer (up to solver tolerance).

**Proposition 2** (Well-posedness and convergence (informal)). *For convex $f$, the proximal objective*

$$\phi_t(\boldsymbol{x}) = f(x) + \frac{1}{2\eta}\|\boldsymbol{x} - \boldsymbol{x}_t\|_2^2$$

*is $(1/\eta)$-strongly convex and admits a unique minimizer $\boldsymbol{x}_{t+1} = \mathrm{prox}_{\eta f}(\boldsymbol{x}_t)$.*

*On a bounded sublevel set around $x_t$, say contained in a ball $\overline{B}(\boldsymbol{x}_t, R)$ of radius $R$, the gradient $\nabla f$ has a local Lipschitz constant $L_2(f, R)$. In this region, $\nabla\phi_t$ is Lipschitz with constant $L_2(\phi, R) = L_2(f, R) + 1/\eta$, and gradient descent with either fixed stepsizes $\lambda \leq 1/L_2(\phi, R)$ or Armijo backtracking (Armijo, 1966) converges linearly. The rate is controlled by the local condition number $\kappa(R) = 1 + \eta L_2(f, R)$.*

**Practical solver.** In practice, we approximate $\mathrm{prox}_{\eta f}(\boldsymbol{x}_t)$ via gradient descent with Armijo backtracking, summarized in Algorithm 1. This ensures convergence to arbitrary precision, so the 1-Lipschitz property of $\mathrm{prox}_{\eta f}$ carries over to the implemented layer (up to solver tolerance).

---

**Algorithm 1** Gradient descent with Armijo backtracking for $(P_t)$

---

**Require:** $\boldsymbol{x}_t \in \mathbb{R}^{Nd}$, tolerance $\varepsilon > 0$
1: $\boldsymbol{y}^{(0)} \leftarrow \boldsymbol{x}_t$
2: **for** $k = 0, 1, 2, \ldots$ **do**
3: $\quad \boldsymbol{g}^{(k)} \leftarrow \nabla f(\boldsymbol{y}^{(k)}) + \frac{1}{\eta}(\boldsymbol{y}^{(k)} - \boldsymbol{x}_t)$
4: $\quad$ **if** $\|\boldsymbol{g}^{(k)}\|_2 \leq \varepsilon$ **then**
5: $\quad\quad$ **break**
6: $\quad$ **end if**
7: $\quad$ Choose $\lambda_k$ by Armijo backtracking
8: $\quad \boldsymbol{y}^{(k+1)} \leftarrow \boldsymbol{y}^{(k)} - \lambda_k \boldsymbol{g}^{(k)}$
9: **end for**
10: $\boldsymbol{x}_{t+1} \leftarrow \boldsymbol{y}^{(k)}$

---

Armijo backtracking (see Algorithm 2) requires an initial step guess. In practice, setting $\lambda_{\mathrm{init}}$ from an estimate of the local Lipschitz constant $L$ provides a good starting point and often accelerates convergence. Note that the inner iteration of Armijo does not require gradient accumulation.

**Role of the step parameter.** A central advantage of the implicit (proximal) scheme is that $\eta > 0$ can be chosen without restriction: for any value, $\mathrm{prox}_{\eta f}$ remains firmly non-expansive and therefore 1-Lipschitz. In practice, $\eta$ serves as a tunable hyperparameter: larger values increase the influence of $f$, while smaller values improve the conditioning of the subproblem. The choice of $\eta$ directly impacts the local condition number $\kappa(R) = 1 + \eta L_2(f, R)$, and can thus be guided by local Lipschitz estimates of $\nabla f$ (see Appendix A.3.2).

**From dynamical flow to block.** Our construction starts from a continuous-time gradient flow derived from a convex potential and stabilizes it via an implicit Euler step. Reformulating this step as a proximal operator yields a tractable update rule that is contractive by design. At the discrete level, this update defines the actual attention block, taking an input $\boldsymbol{x}_t$ and producing an output $\boldsymbol{x}_{t+1}$.

**Theorem 4** (Multi-head convex attention potential). *For each head $h = 1, \ldots, H$, define the convex potential*

$$f^h(\boldsymbol{x}) = \frac{1}{2}\sum_{i=1}^{N} \mathrm{LogSumExp}\big(\boldsymbol{z}_i^h(\boldsymbol{x})\big), \qquad \boldsymbol{z}_i^h(\boldsymbol{x}) = \Big\{ S_{ij}^{convex,h}(\boldsymbol{x}) \Big\}_{j=1}^{N},$$

*and let $f(\boldsymbol{x}) = \sum_{h=1}^{H} f^h(\boldsymbol{x})$. Then the implicit update*

$$\mathrm{AttLip}_\eta(\boldsymbol{x}_t) = \mathrm{prox}_{\eta f}(\boldsymbol{x}_t)$$

*defines the multi-head convex attention block. The resulting mapping $x_t \mapsto \mathrm{AttLip}_\eta(x_t) = x_{t+1}$ is firmly non-expansive and therefore 1-Lipschitz.*

Thus, the block we implement is both faithful to attention structure and globally 1-Lipschitz—unlike ad-hoc clipping or normalization, which retain sequence-length dependence, while remaining independent of sequence length or parameter norms.

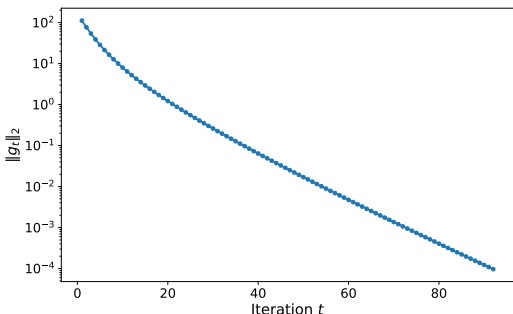

Figure 1: Convergence speed of the inner gradient descent solver for the proximal subproblem ($P_t$). We report $\|g^{(t)}\|_2$ as a function of inner iteration $t$ for $N = 252$, embedding dimension $d = 512$, and a single head ($H = 1$). Although the proximal objective is convex but not smooth, the residual decreases rapidly within a few iterations.

| Layer | Metric | $N=16$ | $N=32$ | $N=64$ | $N=128$ | $N=256$ | $N=512$ | $N=1024$ | $N=2048$ |
|---|---|---|---|---|---|---|---|---|---|
| AttLip (ours) | PGD | 0.991 | 0.991 | 0.990 | 0.990 | 0.989 | 0.987 | 0.987 | 0.987 |
| | Theoretical | 1 | 1 | 1 | 1 | 1 | 1 | 1 | 1 |
| $\text{Att}_{\ell_2}$ | PGD | 2.521 | 2.480 | 2.558 | 2.551 | 2.513 | 2.622 | 2.455 | 2.518 |
| | Theoretical | 28.7 | 52.6 | 90.9 | 153.7 | 253.1 | 407.7 | 662.5 | 1055.5 |

Table 1: Comparison between our AttLip (ours) block and the $\text{Att}_{\ell_2}$ layer of Kim et al. (2021), with embedding dimension $d = 512$ and number of heads $H = 8$. We report the empirical Lipschitz constant estimated by projected gradient ascent (PGD) and the corresponding theoretical upper bound. Here $N$ denotes the input sequence length. We run the solver for $K = 20$ iterations in our layer to ensure convergence. Our bound is fixed to 1, independent of $N$, while $\text{Att}_{\ell_2}$ grows as $O(N \log N)$ and becomes rapidly vacuous.

## 5 EXPERIMENTS

In all experiments, we arbitrarily fix $\eta = 1$, and for the Armijo rule we use standard $\beta = 0.5$, $c = 10^{-4}$, with at most 20 backtracking steps.

### 5.1 THEORETICAL VALIDATIONS

**Speed of convergence of proximal problem** Figure 1 shows that the solver (Algorithm 1 used) converges quickly even without smoothness assumptions. While general convex non-smooth optimization admits only $O(1/t)$ rates, the structure of $\phi_k$ leads to much faster empirical decay, keeping the overhead of the proximal formulation modest in practice. Empirically, the convergence rate seems independent of $N$, so the cost of computing the proximal update remains stable as sequence length grows.

**Comparison with previous Lipschitz attention** Table 1 highlights the difference between our AttLip and the $\text{Att}_{\ell_2}$ layer of Kim et al. (2021). All layers are initialized with the standard initialization used in Transformer architectures (Gaussian weights with variance scaling, as in Vaswani et al. (2017)). A key advantage of our construction is that the theoretical Lipschitz bound is *independent of the sequence length $N$* and remains fixed at 1 by design. This contrasts with $\text{Att}_{\ell_2}$, whose theoretical bound grows linearly with $N$ and quickly becomes vacuous. Moreover, our theoretical bound is much tighter and consistently matches the empirical estimates obtained with adversarial search ("PGD max"), where we maximize the local Lipschitz ratio by projected gradient ascent within an $\ell_2$-ball around the input (Shi et al., 2022), while $\text{Att}_{\ell_2}$ exhibits a large gap between its loose theoretical guarantee and its empirical Lipschitz behavior. In large-scale foundation models such as LLaMA–65B (Touvron et al., 2023), the Transformer stack contains up to 80 layers. In such

regimes, estimating the global Lipschitz constant quickly becomes computationally prohibitive, and the resulting bound can be extremely large and thus uninformative.

These results confirm that our bound is not only formally tight but also practically verifiable, which is a prerequisite before testing whether such stability constraints still permit non-trivial learning capacity.

## 5.2 ONE-LAYER EXPERIMENTS ON OPENWEBTEXT

We train on the OpenWebText corpus in the non-causal next-token prediction setting, following the preprocessing and hyperparameters of the original RoBERTa paper (Liu et al., 2019). As a first step, we restrict to small one-layer networks in order to isolate the role of attention. Specifically, we compare four variants: (i) a baseline with standard dot-product attention, (ii) $\text{Att}_{\ell_2}$ from Kim et al. (2021), (iii) a one-layer RoBERTa model without attention (MLP head only), and (iv) our proposed block $\text{AttLip}_\eta$, which enforces global 1-Lipschitzness. This setup allows us to test whether token-mixing through different attention mechanisms improves learning beyond feedforward-only architectures.

For our method, we denote by $K_{\text{prox}}$ the number of inner gradient steps used to approximately solve the proximal subproblem in each layer, and by $K_{\text{armijo}}$ the iterations of Armijo backtracking. Since Armijo steps do not accumulate gradients, their cost is negligible compared to $K_{\text{prox}}$. In practice, we train with very few proximal steps (typically $K_{\text{prox}} = 1$–3) and use larger values only at inference. To further reduce cost, we adopt a simple schedule where training begins with a single step and gradually increases to two or more. This keeps the overhead modest while ensuring strict 1-Lipschitzness at inference.

| Model | Lipschitz constant | Val. PPL | PPL $\times$ Lip |
|---|---|---|---|
| MLP (no attention) | — | 988 | — |
| Standard attention | unbounded | 22 | — |
| $\text{Att}_{\ell_2}$ (Kim et al., 2021) | 2117 | 28 | $5.9 \times 10^4$ |
| Our block $\text{AttLip}$ | 1 | 141 | 141 |

Table 2: Validation perplexity and Lipschitz constants on OpenWebText (1-layer models). Both metrics should be minimized. The product (PPL $\times$ Lip) provides a simple joint measure of efficiency and robustness: **lower values are better**.

The results in Table 2 highlight the trade-off between expressivity and contractivity. Our block enforces global 1-Lipschitzness by design, a qualitatively different guarantee than existing mechanisms. Although its validation perplexity (141) is higher than that of $\text{Att}_{\ell_2}$ (28) or standard attention (22), the combined score (PPL $\times$ Lip) provides a simple proxy measure for robust learning and shows that $\text{AttLip}$ is the only mechanism achieving non-trivial modeling ability under exact Lipschitz control (141 vs. $5.9 \times 10^4$ for $\text{Att}_{\ell_2}$). This demonstrates that non-trivial token interactions can be captured while maintaining strict robustness guarantees. Standard attention attains the best perplexity but at the cost of being non-Lipschitz, whereas $\text{Att}_{\ell_2}$ sacrifices robustness entirely with a very large constant.

## 6 CONCLUSION

We introduced a novel self-attention mechanism based on convex proximal flows, which guarantees global 1-Lipschitz continuity by construction. Our approach provides a principled alternative to previous Lipschitz attention variants, combining operator-theoretic contractivity with efficient approximate solvers. The experiment on OpenWebText demonstrates that, despite operating under strict Lipschitz constraints, the proposed block can capture non-trivial token dependences.

The present work establishes a theoretical and algorithmic foundation for Lipschitz attention in large-scale architectures. Future work should investigate scaling to deeper models and applying the framework to downstream tasks where robustness guarantees are critical. In particular, extending the analysis to adversarial perturbations in the embedding space for NLP remains an open direction.

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

# A  APPENDIX

## A.1  FINITE-$K$ APPROXIMATION AND THE ROLE OF $\eta$

In practice, the implicit convex-attention update is computed by running only a finite number $K$ of iterations of the inner solver rather than solving the proximal subproblem exactly. The behaviour of this truncated update is governed by the proximal parameter $\eta$, which induces a natural tradeoff between convergence speed and expressivity. When $\eta$ is small, the quadratic term $(2\eta)^{-1}\|\boldsymbol{x} - \boldsymbol{x}_t\|^2$ strongly penalizes deviations from the input, so the update remains close to the identity map; the subproblem is then strongly convex and the inner iterations converge quickly. Increasing $\eta$ weakens this quadratic regularization, allowing the update to move further from $\boldsymbol{x}_t$ and thus yielding a more expressive transformation, but at the cost of slower inner convergence due to reduced strong convexity. The next proposition shows that the resulting finite-$K$ update remains close to the exact 1-Lipschitz proximal map, with an explicit error term controlled by the solver's residual.

In the following, let $f : \mathbb{R}^d \to \mathbb{R}$ be the convex potential corresponding to the multi-head convex attention block of Theorem 4, after reshaping $\boldsymbol{x} \in \mathbb{R}^{N \times d_{\mathrm{model}}}$ into a vector in $\mathbb{R}^d$ with $d = N d_{\mathrm{model}}$.

**Proposition 3** (Approximate convex attention). *Fix $\eta > 0$. For each input $\boldsymbol{x}_t \in \mathbb{R}^d$, define*

$$\varphi_{\boldsymbol{x}_t}(\boldsymbol{x}) := f(\boldsymbol{x}) + \frac{1}{2\eta}\|\boldsymbol{x} - \boldsymbol{x}_t\|_2^2,$$

*and denote the exact proximal update by*

$$\mathrm{AttLip}_\eta(\boldsymbol{x}_t) := \operatorname*{prox}_{\eta f}(\boldsymbol{x}_t) = \arg\min_{\boldsymbol{x} \in \mathbb{R}^d} \varphi_{\boldsymbol{x}_t}(\boldsymbol{x}).$$

*Assume that, instead of $\mathrm{AttLip}_\eta(\boldsymbol{x}_t)$, the block implements an approximate solution $\mathrm{AttLip}_\eta^{(K)}(\boldsymbol{x}_t)$ obtained after $K$ inner iterations of a solver, and define the gradient residual*

$$\boldsymbol{g}_K(\boldsymbol{x}_t) := \nabla_{\boldsymbol{x}} \varphi_{\boldsymbol{x}_t}\big(\mathrm{AttLip}_\eta^{(K)}(\boldsymbol{x}_t)\big) \in \mathbb{R}^d.$$

*Suppose there exists a function $\varepsilon : \mathbb{R}^d \to [0, +\infty)$ such that*

$$\|\boldsymbol{g}_K(\boldsymbol{x}_t)\|_2 \leq \varepsilon(\boldsymbol{x}_t) \qquad \text{for all } \boldsymbol{x}_t \in \mathbb{R}^d.$$

*Then:*

1. *(Pointwise approximation of the exact convex attention block) for all $\boldsymbol{x}_t \in \mathbb{R}^d$,*

$$\|\mathrm{AttLip}_\eta^{(K)}(\boldsymbol{x}_t) - \mathrm{AttLip}_\eta(\boldsymbol{x}_t)\|_2 \leq \eta\,\varepsilon(\boldsymbol{x}_t).$$

2. *(Almost 1-Lipschitz behaviour) for all $\boldsymbol{x}_t, \boldsymbol{y}_t \in \mathbb{R}^d$,*

$$\|\mathrm{AttLip}_\eta^{(K)}(\boldsymbol{x}_t) - \mathrm{AttLip}_\eta^{(K)}(\boldsymbol{y}_t)\|_2 \leq \|\boldsymbol{x}_t - \boldsymbol{y}_t\|_2 + \eta\big(\varepsilon(\boldsymbol{x}_t) + \varepsilon(\boldsymbol{y}_t)\big).$$

   *In particular, if $\varepsilon(\boldsymbol{x}_t) \leq \varepsilon$ for all $\boldsymbol{x}_t$, then*

$$\|\mathrm{AttLip}_\eta^{(K)}(\boldsymbol{x}_t) - \mathrm{AttLip}_\eta^{(K)}(\boldsymbol{y}_t)\|_2 \leq \|\boldsymbol{x}_t - \boldsymbol{y}_t\|_2 + 2\,\eta\,\varepsilon \qquad \forall\,\boldsymbol{x}_t, \boldsymbol{y}_t \in \mathbb{R}^d.$$

*Proof.* Fix $\boldsymbol{x}_t \in \mathbb{R}^d$. The function

$$\varphi_{\boldsymbol{x}_t}(\boldsymbol{x}) = f(\boldsymbol{x}) + \tfrac{1}{2\eta}\|\boldsymbol{x} - \boldsymbol{x}_t\|_2^2$$

is $\mu$-strongly convex with $\mu = 1/\eta$, hence it has a unique minimizer, denoted $\boldsymbol{x}^\star(\boldsymbol{x}_t) := \mathrm{AttLip}_\eta(\boldsymbol{x}_t)$.

Strong convexity yields, for all $\boldsymbol{x} \in \mathbb{R}^d$,

$$\big\|\nabla \varphi_{\boldsymbol{x}_t}(\boldsymbol{x}) - \nabla \varphi_{\boldsymbol{x}_t}(\boldsymbol{x}^\star(\boldsymbol{x}_t))\big\|_2 \geq \mu\,\|\boldsymbol{x} - \boldsymbol{x}^\star(\boldsymbol{x}_t)\|_2.$$

Since $\nabla \varphi_{\boldsymbol{x}_t}(\boldsymbol{x}^\star(\boldsymbol{x}_t)) = \boldsymbol{0}$, we obtain

$$\|\nabla \varphi_{\boldsymbol{x}_t}(\boldsymbol{x})\|_2 \geq \mu\,\|\boldsymbol{x} - \boldsymbol{x}^\star(\boldsymbol{x}_t)\|_2.$$

Applying this to $\boldsymbol{x} = \mathrm{AttLip}_\eta^{(K)}(\boldsymbol{x}_t)$ gives

$$\|\mathrm{AttLip}_\eta^{(K)}(\boldsymbol{x}_t) - \mathrm{AttLip}_\eta(\boldsymbol{x}_t)\|_2 \;\leq\; \frac{1}{\mu}\, \|\nabla\varphi_{\boldsymbol{x}_t}(\mathrm{AttLip}_\eta^{(K)}(\boldsymbol{x}_t))\|_2 = \eta\,\|\boldsymbol{g}_K(\boldsymbol{x}_t)\|_2 \;\leq\; \eta\,\varepsilon(\boldsymbol{x}_t),$$

which proves item (1).

For item (2), recall that $\mathrm{AttLip}_\eta = \mathrm{prox}_{\eta f}$ is firmly non-expansive in $(\mathbb{R}^d, \|\cdot\|_2)$, hence

$$\|\mathrm{AttLip}_\eta(\boldsymbol{x}_t) - \mathrm{AttLip}_\eta(\boldsymbol{y}_t)\|_2 \;\leq\; \|\boldsymbol{x}_t - \boldsymbol{y}_t\|_2.$$

For arbitrary $\boldsymbol{x}_t, \boldsymbol{y}_t$,

$$\|\mathrm{AttLip}_\eta^{(K)}(\boldsymbol{x}_t) - \mathrm{AttLip}_\eta^{(K)}(\boldsymbol{y}_t)\|_2 \leq \|\mathrm{AttLip}_\eta(\boldsymbol{x}_t) - \mathrm{AttLip}_\eta(\boldsymbol{y}_t)\|_2 + \|\mathrm{AttLip}_\eta^{(K)}(\boldsymbol{x}_t) - \mathrm{AttLip}_\eta(\boldsymbol{x}_t)\|_2$$
$$+ \|\mathrm{AttLip}_\eta^{(K)}(\boldsymbol{y}_t) - \mathrm{AttLip}_\eta(\boldsymbol{y}_t)\|_2$$
$$\leq \|\boldsymbol{x}_t - \boldsymbol{y}_t\|_2 + \eta\,\varepsilon(\boldsymbol{x}_t) + \eta\,\varepsilon(\boldsymbol{y}_t),$$

using item (1) for both $\boldsymbol{x}_t$ and $\boldsymbol{y}_t$. If $\varepsilon(\boldsymbol{x}_t) \leq \varepsilon$ for all $\boldsymbol{x}_t$, then

$$\|\mathrm{AttLip}_\eta^{(K)}(\boldsymbol{x}_t) - \mathrm{AttLip}_\eta^{(K)}(\boldsymbol{y}_t)\|_2 \;\leq\; \|\boldsymbol{x}_t - \boldsymbol{y}_t\|_2 + 2\,\eta\,\varepsilon,$$

completing the proof. $\qquad\square$

**Composition of approximate proximal layers.** If several finite-iteration proximal blocks are stacked, each satisfying an almost non-expansive bound of the form $\|F_\ell(u) - F_\ell(v)\|_2 \leq \|u - v\|_2 + 2\eta_\ell\varepsilon_\ell$, then their composition also remains almost 1-Lipschitz, with a combined additive defect equal to the sum of the individual ones:

$$\|F_L \circ \cdots \circ F_1(u) - F_L \circ \cdots \circ F_1(v)\|_2 \;\leq\; \|u - v\|_2 + 2\sum_{\ell=1}^{L} \eta_\ell\varepsilon_\ell.$$

Thus, a deep stack of such implicit layers preserves stability, and the Lipschitz degradation accumulates in a transparent and explicit controlled way.

A.2 ALGORITHMS

---

**Algorithm 2** Backtracking Armijo rule for $\phi_t$

---
**Require:** $0 < \beta < 1$, $c \in (0, 1)$
  1: $\lambda \leftarrow \lambda_{\mathrm{init}}$                                                            (e.g. from local $L$ estimate)
  2: **while** $\phi_t\big(\boldsymbol{y}^{(k)} - \lambda\boldsymbol{g}^{(k)}\big) > \phi_t\big(\boldsymbol{y}^{(k)}\big) - c\lambda\|\boldsymbol{g}^{(k)}\|_2^2$ **do**
  3:     $\lambda \leftarrow \beta\lambda$
  4: **end while**
  5: **return** $\lambda$

---

In practice, we use standard Armijo (Algorithm 2) parameters with backtracking factor $\beta = 0.5$, slope constant $c = 10^{-4}$, and at most 20 backtracking steps.

---

**Algorithm 3** Standard multi-head attention

---
**Require:** Input $\boldsymbol{X} \in \mathbb{R}^{N \times d}$
  1: **for** $h = 1, \ldots, H$ **do**
  2:     $\boldsymbol{Q}^h \leftarrow \boldsymbol{X}\boldsymbol{W}_{Q,h}^\top, \quad \boldsymbol{K}^h \leftarrow \boldsymbol{X}\boldsymbol{W}_{K,h}^\top, \quad \boldsymbol{V}^h \leftarrow \boldsymbol{X}\boldsymbol{W}_{V,h}^\top$
  3:     $\boldsymbol{S}^h \leftarrow \frac{1}{\sqrt{d_h}}\boldsymbol{Q}^h\boldsymbol{K}^{h\top}$
  4:     $\boldsymbol{A}^h \leftarrow \mathrm{softmax}_{\mathrm{row}}(\boldsymbol{S}^h)$
  5:     $\boldsymbol{Y}^h \leftarrow \boldsymbol{A}^h\boldsymbol{V}^h$
  6: **end for**
  7: **return** $\mathrm{concat}_{h=1}^{H}\boldsymbol{Y}_h\,\boldsymbol{W}^O$

---

---

**Algorithm 4** Gradient of convex attention potential $\nabla f(\boldsymbol{X})$

---

**Require:** Input $\boldsymbol{X} \in \mathbb{R}^{N \times d}$, head projections $\{\boldsymbol{W}^h\}_{h=1}^H$
1: **for** $h = 1, \dots, H$ **do**
2:    $\boldsymbol{V}^h \leftarrow \boldsymbol{X}(\boldsymbol{W}^h)^\top \quad \in \mathbb{R}^{N \times d_h}$
3:    $\mathbf{u}^h \leftarrow \|\boldsymbol{V}^h\|_2^2 \quad \in \mathbb{R}^N$              (rowwise squared norms: $(\mathbf{u}^h)_i = \|\boldsymbol{V}_i^h\|_2^2$)
4:    $\boldsymbol{S}^h \leftarrow \frac{2}{\sqrt{d_h}} \boldsymbol{V}^h(\boldsymbol{V}^h)^\top + \frac{1}{\sqrt{d_h}}\big(\mathbf{u}^h\mathbf{1}^\top + \mathbf{1}(\mathbf{u}^h)^\top\big)$         $(S_{ij}^h =$
    $\frac{1}{\sqrt{d_h}}(\|V_i^h\|^2 + \|V_j^h\|^2 + 2(V_i^h)^\top V_j^h))$
5:    $\boldsymbol{S}^h \leftarrow \boldsymbol{S}^h - \mathrm{rowmax}(\boldsymbol{S}^h)$                   (stability, row-wise)
6:    $\boldsymbol{A}^h \leftarrow \mathrm{softmax}_{\mathrm{row}}(\boldsymbol{S}^h)$
7:    $\boldsymbol{A}^h \leftarrow \boldsymbol{A}^h + (\boldsymbol{A}^h)^\top$               (adds column contributions)
8:    $\boldsymbol{D}^h \leftarrow \mathrm{diag}\big((\boldsymbol{A}^h)^\top \mathbf{1}\big) + \boldsymbol{I}_N$         (self term $1 + \sum_i \alpha_{i,k}^h$)
9:    $\boldsymbol{Y}^h \leftarrow (\boldsymbol{A}^h + \boldsymbol{D}^h)\boldsymbol{V}^h \quad \in \mathbb{R}^{N \times d_h}$
10: **end for**
11: **return** $\sum_{h=1}^H \boldsymbol{Y}^h \boldsymbol{W}^h \quad \in \mathbb{R}^{N \times d}$

---

### A.3 PROOFS

#### A.3.1 PROOF OF PROPOSITION 4

**Lemma 1** (Convexity of the attention potential). *Let*

$$f(\boldsymbol{x}) = \frac{1}{2}\sum_{i=1}^N \mathrm{LogSumExp}\big(\boldsymbol{z}_i(\boldsymbol{x})\big), \qquad \boldsymbol{z}_i(\boldsymbol{x}) = \big\{(\boldsymbol{x}_i + \boldsymbol{x}_j)^\top \boldsymbol{W}^\top \boldsymbol{W}(\boldsymbol{x}_i + \boldsymbol{x}_j)\big\}_{j=1}^N,$$

*with $\boldsymbol{x} = [\boldsymbol{x}_1^\top \dots \boldsymbol{x}_N^\top]^\top \in \mathbb{R}^{Nd}$, $\boldsymbol{W} \in \mathbb{R}^{d_{\mathrm{head}} \times d}$ and $\boldsymbol{W}^\top \boldsymbol{W} \succeq 0$. Then $f$ is convex over $\mathbb{R}^{Nd}$.*

*Proof.* Fix $i$ and $j$. Write

$$z_{i,j}(\boldsymbol{x}) = (\boldsymbol{x}_i + \boldsymbol{x}_j)^\top \boldsymbol{W}^\top \boldsymbol{W}(\boldsymbol{x}_i + \boldsymbol{x}_j) = \|\boldsymbol{W}(\boldsymbol{x}_i + \boldsymbol{x}_j)\|_2^2.$$

Let $\boldsymbol{e}_i, \boldsymbol{e}_j \in \mathbb{R}^N$ be the canonical vectors and define the linear map $T_{i,j} : \mathbb{R}^{Nd} \to \mathbb{R}^d$ by $T_{i,j}\boldsymbol{x} = \boldsymbol{W}\big((\boldsymbol{e}_i^\top \otimes \boldsymbol{I}_d)\boldsymbol{x} + (\boldsymbol{e}_j^\top \otimes \boldsymbol{I}_d)\boldsymbol{x}\big)$. Then $z_{i,j}(\boldsymbol{x}) = \|T_{i,j}\boldsymbol{x}\|_2^2 = (T_{i,j}\boldsymbol{x})^\top (T_{i,j}\boldsymbol{x}) = \boldsymbol{x}^\top T_{i,j}^\top T_{i,j}\boldsymbol{x}$, a quadratic form with PSD matrix $T_{i,j}^\top T_{i,j} \succeq 0$, hence $z_{i,j}$ is convex. Therefore the vector $\boldsymbol{z}_i(\boldsymbol{x}) = (z_{i,1}(\boldsymbol{x}), \dots, z_{i,N}(\boldsymbol{x}))$ has convex components.

The map $\mathrm{LogSumExp} : \mathbb{R}^N \to \mathbb{R}$, defined by $\mathrm{LogSumExp}(\boldsymbol{u}) = \log\sum_{k=1}^N e^{u_k}$, is convex and (componentwise) nondecreasing. By the monotone composition rule, $\mathrm{LogSumExp}(\boldsymbol{z}_i(\boldsymbol{x}))$ is convex as a composition of a convex, nondecreasing function with a vector of convex functions. Summing over $i$ and multiplying by $1/2$ preserves convexity, hence $f$ is convex. $\square$

*Proof.* By Lemma 1, $f$ is convex and differentiable, so $\nabla f$ is a monotone operator. Applying the contractivity result for gradient systems driven by convex potentials (Theorem 3) to $F = -\nabla f$ yields the claimed nonexpansiveness in the Euclidean norm. $\square$

#### A.3.2 PROOF OF PROPOSITION 2

First, let us restate Proposition 2 more formally:

**Proposition 4** (Well-posedness and conditioning of the proximal subproblem (formal)). *Let $N, d, H \in \mathbb{N}_{\geq 1}$, identify $(\mathbb{R}^d)^N \simeq \mathbb{R}^{Nd}$, and fix PSD matrices $\boldsymbol{A}^h \succeq 0$ for $h = 1, \dots, H$. For $\boldsymbol{x} = (\boldsymbol{x}_1, \dots, \boldsymbol{x}_N) \in \mathbb{R}^{Nd}$ with $\boldsymbol{x}_i \in \mathbb{R}^d$, define the scores*

$$S_{i,j}^h(\boldsymbol{x}) := (\boldsymbol{x}_i + \boldsymbol{x}_j)^\top \boldsymbol{A}^h(\boldsymbol{x}_i + \boldsymbol{x}_j), \qquad f(\boldsymbol{x}) := \frac{1}{2}\sum_{h=1}^H \sum_{i=1}^N \mathrm{LogSumExp}\big(S_{i,1}^h(\boldsymbol{x}), \dots, S_{i,N}^h(\boldsymbol{x})\big).$$

*Fix $\eta > 0$ and $\boldsymbol{x}_t \in \mathbb{R}^{Nd}$, and consider*

$$\phi_t(\boldsymbol{x}) := f(\boldsymbol{x}) + \frac{1}{2\eta}\|\boldsymbol{x} - \boldsymbol{x}_t\|^2, \qquad \boldsymbol{x}_{t+1} := \arg\min_{\boldsymbol{x} \in \mathbb{R}^{Nd}} \phi_t(\boldsymbol{x}).$$

*Let the sublevel set $D := \{\boldsymbol{x} : \phi_t(\boldsymbol{x}) \leq \phi_t(\boldsymbol{x}_t)\}$ and choose $R > 0$ with $D \subset \overline{B}(\boldsymbol{x}_t, R)$. Define the local smoothness constant*

$$L_2(f, R) := \sup_{\boldsymbol{x} \in \overline{B}(\boldsymbol{x}_t, R)} \|\nabla^2 f(\boldsymbol{x})\| < \infty,$$

*then set $L_2(\phi, R) := L_2(f, R) + 1/\eta$ and the local condition number $\kappa(R) := 1 + \eta L_2(f, R)$. Then:*

1. ***Strong convexity, uniqueness, proximal characterization.*** *$\phi_t$ is $(1/\eta)$-strongly convex on $\mathbb{R}^{Nd}$, hence admits a unique minimizer $\boldsymbol{x}_{t+1}$. Moreover $\boldsymbol{x}_{t+1} = \mathrm{prox}_{\eta f}(\boldsymbol{x}_t) = (I + \eta \, \partial f)^{-1}(\boldsymbol{x}_t) = (I + \eta \, \nabla f)^{-1}(\boldsymbol{x}_t)$.*

2. ***Bounded sublevel set and local smoothness.*** *$D$ is nonempty, closed, and bounded. Furthermore, $f \in C^2$, so $\nabla^2 f$ is continuous and $L_2(f, R) < \infty$. Consequently, $\nabla f$ is $L_2(f, R)$-Lipschitz on $\overline{B}(\boldsymbol{x}_t, R)$ and $\nabla \phi_t$ is $L_2(\phi, R)$-Lipschitz on $\overline{B}(\boldsymbol{x}_t, R)$.*

3. ***Gradient descent with guaranteed containment and linear rate.*** *Let $\boldsymbol{y}^{(0)} := \boldsymbol{x}_t$ and, for any $\lambda \in \left(0, 1/L_2(\phi, R)\right]$, define $\boldsymbol{y}^{(k+1)} := \boldsymbol{y}^{(k)} - \lambda \nabla \phi_t(\boldsymbol{y}^{(k)})$. Then $\boldsymbol{y}^{(k)} \in D \subset \overline{B}(\boldsymbol{x}_t, R)$ for all $k$, and*

$$\|\boldsymbol{y}^{(k)} - \boldsymbol{x}_{t+1}\| \leq \left(1 - \frac{1}{\kappa(R)}\right)^k \|\boldsymbol{x}_t - \boldsymbol{x}_{t+1}\|, \qquad k = 0, 1, 2, \ldots.$$

4. ***Armijo backtracking: step lower bound and linear convergence.*** *Fix $\sigma \in (0, 1/2)$ and $\beta \in (0, 1)$. At iteration $k$, choose $\lambda_k$ by backtracking from any trial step, shrinking by $\beta$ until*

$$\phi_t\big(\boldsymbol{y}^{(k)} - \lambda_k \nabla \phi_t(\boldsymbol{y}^{(k)})\big) \leq \phi_t(\boldsymbol{y}^{(k)}) - \sigma \lambda_k \|\nabla \phi_t(\boldsymbol{y}^{(k)})\|^2.$$

*Then $\lambda_k \geq \beta(1 - \sigma)/L_2(\phi, R)$ for all $k$, $\boldsymbol{y}^{(k)} \in D$ for all $k$, and there exists $c = c(\beta, \sigma) \in (0, 1)$ such that*

$$\|\boldsymbol{y}^{(k)} - \boldsymbol{x}_{t+1}\| \leq \left(1 - \frac{c}{\kappa(R)}\right)^k \|\boldsymbol{x}_t - \boldsymbol{x}_{t+1}\| \qquad \text{for all } k \in \mathbb{N}.$$

*Proof.* **(1) Strong convexity and proximal map.** Each $S_{i,j}^h$ is convex quadratic since $\boldsymbol{A}^h \succeq 0$. The map LogSumExp is convex and coordinatewise nondecreasing; thus $\mathrm{LogSumExp} \circ (S_{i,1}^h, \ldots, S_{i,N}^h)$ is convex, and so is $f$ (finite everywhere). The map $\boldsymbol{x} \mapsto \frac{1}{2\eta}\|\boldsymbol{x} - \boldsymbol{x}_k\|^2$ is $(1/\eta)$-strongly convex, hence $\phi_t = f + \frac{1}{2\eta}\| \cdot - \boldsymbol{x}_t\|^2$ is $(1/\eta)$-strongly convex and admits a unique minimizer $\boldsymbol{x}_{t+1}$. The first-order optimality condition $0 \in \partial f(\boldsymbol{x}_{t+1}) + \frac{1}{\eta}(\boldsymbol{x}_{t+1} - \boldsymbol{x}_t)$ reads $\boldsymbol{x}_{t+1} = (\mathrm{Id} + \eta \, \partial f)^{-1}(\boldsymbol{x}_t)$; since $f$ is $C^1$ convex, $\partial f = \{\nabla f\}$ and $\boldsymbol{x}_{t+1} = (I + \eta \, \nabla f)^{-1}(\boldsymbol{x}_t) = \mathrm{prox}_{\eta f}(\boldsymbol{x}_t)$.

**(2) Bounded sublevel set; local smoothness.** Nonemptiness: $\boldsymbol{x}_t \in D$ by definition. Closedness: $\phi_t$ is continuous. To show boundedness, note $S_{i,j}^{(h)}(x) \geq 0$ for all $x$ and $(h, i, j)$, hence $\mathrm{LogSumExp}(S_{i,1}^h(\boldsymbol{x}), \ldots, S_{i,N}^h(\boldsymbol{x})) \geq \log\big(\sum_{j=1}^N e^0\big) = \log N$. Therefore $f(\boldsymbol{x}) \geq \frac{1}{2} HN \log N$, and

$$\phi_t(\boldsymbol{x}) \geq \frac{1}{2\eta}\|\boldsymbol{x} - \boldsymbol{x}_t\|^2 + \frac{1}{2} HN \log N \xrightarrow[\|\boldsymbol{x}\| \to \infty]{} \infty,$$

so $\phi_t$ is coercive and $D$ is bounded. Since $f$ is a finite sum of $C^\infty$ functions, $f \in C^2$ and $\nabla^2 f$ is continuous. Hence $L_2(f, R) := \sup_{\boldsymbol{x} \in \overline{B}(\boldsymbol{x}_t, R)} \|\nabla^2 f(\boldsymbol{x})\| < \infty$ and $\nabla f$ is $L_2(f, R)$-Lipschitz on $\overline{B}(\boldsymbol{x}_t, R)$. Adding the quadratic gives that $\nabla \phi_t$ is $L_2(\phi, R) = L_2(f, R) + 1/\eta$-Lipschitz on $\overline{B}(\boldsymbol{x}_r, R)$.

**(3) Gradient descent: containment and rate for $\lambda \leq 1/L_2(\phi, R)$.** Let $\mu := 1/\eta$ and $L := L_2(\phi, R)$. Because $\phi_t$ is convex, each sublevel set is convex; in particular, $D$ is convex. On $\overline{B}(x_k, R)$ the *descent lemma* holds: for any $\boldsymbol{u}, \boldsymbol{v}$ with the segment $[\boldsymbol{u}, \boldsymbol{v}] \subset \overline{B}(\boldsymbol{x}_t, R)$,

$$\phi_t(\boldsymbol{v}) \leq \phi_t(\boldsymbol{u}) + \langle \nabla \phi_t(\boldsymbol{u}), \boldsymbol{v} - \boldsymbol{u} \rangle + \frac{L}{2}\|\boldsymbol{v} - \boldsymbol{u}\|^2.$$

Apply it with $\boldsymbol{u} = \boldsymbol{y}^{(k)}$ and $v = y^{(k)} - \lambda\nabla\phi_k(y^{(k)})$. For any $\lambda \in (0, 1/L]$ we obtain

$$\phi_t(\boldsymbol{y}^{(k+1)}) \;\leq\; \phi_t(\boldsymbol{y}^{(k)}) - \left(\lambda - \tfrac{L}{2}\lambda^2\right)\|\nabla\phi_t(\boldsymbol{y}^{(k)})\|^2 \;\leq\; \phi_t(\boldsymbol{y}^{(k)}) - \tfrac{\lambda}{2}\|\nabla\phi_t(\boldsymbol{y}^{(k)})\|^2 \;\leq\; \phi_t(\boldsymbol{y}^{(k)}).$$

By induction, $\phi_t(\boldsymbol{y}^{(k)}) \leq \phi_t(x_t)$ for all $k$, so $y^{(k)} \in D \subset \overline{B}(\boldsymbol{x}_t, R)$. Since $D$ is convex, the segment between $\boldsymbol{y}^{(k)}$ and $\boldsymbol{y}^{(k+1)}$ lies in $D$ and thus in $\overline{B}(\boldsymbol{x}_t, R)$, justifying the use of the descent lemma at every step.

For the linear rate in distance, observe that $\phi_t \in C^2$ and, for all $\boldsymbol{x} \in \overline{B}(x_t, R)$,

$$\mu\mathrm{Id} \;\preceq\; \nabla^2\phi_t(\boldsymbol{x}) \;=\; \nabla^2 f(\boldsymbol{x}) + \tfrac{1}{\eta}\mathrm{Id} \;\preceq\; LI.$$

Let $\boldsymbol{x}_\star := x_{t+1}$ and write $\boldsymbol{e}^{(k)} := \boldsymbol{y}^{(k)} - \boldsymbol{x}_\star$. By the mean value form of the gradient,

$$\boldsymbol{e}^{(k+1)} = \boldsymbol{e}^{(k)} - \lambda\big(\nabla\phi_t(\boldsymbol{y}^{(k)}) - \nabla\phi_t(\boldsymbol{x}_\star)\big) = \Big(\mathrm{Id} - \lambda\,\overline{H}_k\Big)\boldsymbol{e}^{(k)},$$

where $\overline{H}_k := \int_0^1 \nabla^2\phi_t\big(\boldsymbol{x}_\star + \theta\boldsymbol{e}^{(k)}\big)\,d\theta$. Each $\overline{H}_k$ is symmetric with spectrum in $[\mu, L]$, hence

$$\|\boldsymbol{e}^{(k+1)}\| \;\leq\; \|I - \lambda\overline{H}_k\|\,\|\boldsymbol{e}^{(k)}\| \;\leq\; \rho\,\|\boldsymbol{e}^{(k)}\|, \qquad \rho := \max\{|1 - \lambda\mu|,\ |1 - \lambda L|\}.$$

For $\lambda \in (0, 1/L]$ one has $\rho = 1 - \lambda\mu \leq 1 - \mu/L = 1 - 1/\kappa(R)$, so $\|\boldsymbol{e}^{(k)}\| \leq (1 - 1/\kappa(R))^k\|\boldsymbol{e}^{(0)}\|$, as claimed.

**(4) Armijo backtracking.** Fix $\sigma \in (0, 1/2)$ and $\beta \in (0, 1)$. On $\overline{B}(\boldsymbol{x}_t, R)$ the descent lemma implies that, for any $\boldsymbol{y}$ and any $\lambda \leq (1 - \sigma)/L$,

$$\phi_t\big(\boldsymbol{y} - \lambda\nabla\phi_t(\boldsymbol{y})\big) \;\leq\; \phi_t(\boldsymbol{y}) - \lambda\Big(1 - \frac{L\lambda}{2}\Big)\|\nabla\phi_t(\boldsymbol{y})\|^2 \;\leq\; \phi_t(\boldsymbol{y}) - \sigma\lambda\|\nabla\phi_t(\boldsymbol{y})\|^2.$$

Therefore, the backtracking procedure will accept some step $\lambda_k \geq \beta(1 - \sigma)/L$ at each iteration. Monotonicity then yields $\boldsymbol{y}^{(k)} \in L$ for all $t$ and (as above) all segments $[\boldsymbol{y}^{(k)}, \boldsymbol{y}^{(k+1)}] \subset L \subset \overline{B}(\boldsymbol{x}_t, R)$, so the local smoothness bound is valid throughout.

To obtain a linear rate, combine the accepted Armijo decrease with the Polyak–Łojasiewicz inequality valid for $\mu$-strongly convex functions:

$$\phi_t(\boldsymbol{y}^{(k+1)}) - \phi_t(\boldsymbol{x}_\star) \;\leq\; \phi_t(\boldsymbol{y}^{(k)}) - \phi_t(\boldsymbol{x}_\star) - \sigma\lambda_k\|\nabla\phi_t(\boldsymbol{y}^{(k)})\|^2 \;\leq\; \Big(1 - 2\sigma\lambda_k\,\mu\Big)\big(\phi_t(\boldsymbol{y}^{(k)}) - \phi_t(\boldsymbol{x}_\star)\big).$$

Using $\lambda_k \geq \beta(1 - \sigma)/L$ gives $\phi_t(\boldsymbol{y}^{(k)}) - \phi_t(\boldsymbol{x}_\star) \leq \big(1 - c\,\mu/L\big)^t\big(\phi_t(\boldsymbol{x}_t) - \phi_t(\boldsymbol{x}_\star)\big)$ with $c := 2\beta\sigma(1 - \sigma) \in (0, 1)$. By strong convexity, $\|\boldsymbol{y}^{(k)} - \boldsymbol{x}_\star\|^2 \leq \frac{2}{\mu}\big(\phi_t(\boldsymbol{y}^{(k)}) - \phi_t(\boldsymbol{x}_\star)\big)$, which yields the stated distance contraction with factor $1 - c/\kappa(R)$(after adjusting $c$ if desired). $\qquad\square$

**Remark** On steps larger than $1/L_2(\phi, R)$] The classical bound $\rho = \max\{|1 - \lambda\mu|, |1 - \lambda L|\} < 1$ also holds for any $\lambda \in (0, 2/L_2(\phi, R))$ *provided* all iterates remain in $\overline{B}(\boldsymbol{x}_t, R)$ (so that $L$-smoothness applies along the segments used in the mean-value argument). Our statement enforces containment by requiring $\lambda \leq 1/L_2(\phi, R)$ or by using Armijo backtracking.

**Heuristic scaling of the local smoothness.** Write $\boldsymbol{A}^h = \boldsymbol{W}^{h\top}\boldsymbol{W}^h \succeq 0$ and

$$f(\boldsymbol{x}) = \tfrac{1}{2}\sum_{h=1}^{H}\sum_{i=1}^{N}\mathrm{LogSumExp}\Big(S_{i,j}^h(\boldsymbol{x}) : j = 1, \ldots, N\Big), \qquad S_{i,j}^h(\boldsymbol{x}) = (\boldsymbol{x}_i + \boldsymbol{x}_j)^\top\boldsymbol{A}^h(\boldsymbol{x}_i + \boldsymbol{x}_j).$$

For each fixed $(h, i)$, $\nabla_{\boldsymbol{x}}S_{i,j}^{(h)}$ is *linear* in $\boldsymbol{x}$ and supported only on blocks $\boldsymbol{x}_i$ and $\boldsymbol{x}_j$, with $\|\nabla S_{i,j}^h(x)\| \lesssim \|\boldsymbol{A}^h\|\,(\|\boldsymbol{x}_i\| + \|\boldsymbol{x}_j\|)$. On a ball of radius $R$, this gives $\|\nabla S_{i,j}^h(\boldsymbol{x})\| \lesssim \|\boldsymbol{A}^h\|\,R$. The Hessian of LogSumExp composed with $\{S_{i,j}^h\}_j$ splits as a convex combination of the constant Hessians $\nabla^2 S_{i,j}^h$ (each bounded by $\|\boldsymbol{A}^h\|$ in operator norm) plus a covariance term of the gradients, bounded by $\lesssim \|\boldsymbol{A}^h\|^2 R^2$. Summing over $i = 1, \ldots, N$ contributes a linear factor $N$ for each head. Therefore, on $\{\boldsymbol{x} : \|\boldsymbol{x} - \boldsymbol{x}_k\| \leq R\}$ one obtains the local bound

$$\|\nabla^2 f(\boldsymbol{x})\| \lesssim \sum_{h=1}^{H}N\Big(\|\boldsymbol{A}^h\| + \|\boldsymbol{A}^h\|^2 R^2\Big), \qquad\Rightarrow\qquad L_2(f, R) \lesssim \sum_{h=1}^{H}N\Big(\|\boldsymbol{A}^h\| + \|\boldsymbol{A}^h\|^2 R^2\Big).$$

If the weights are spectrally normalized so that $\|\boldsymbol{W}^h\| \leq 1$ (hence $\|\boldsymbol{A}_h\| \leq 1$), this simplifies to

$$L_2(f, R) \lesssim H\,N\,(1 + R^2).$$

Consequently, the local condition number in Proposition **??** behaves as

$$\kappa(R) = 1 + \eta L(f, R) \lesssim 1 + \eta \sum_{h=1}^{H} N\Big(\|\boldsymbol{A}^h\| + \|\boldsymbol{A}^h\|^2 R^2\Big),$$

which is linear in $N$ under spectral normalization.

