# OpenReview forum: "Globally 1-Lipschitz Attention Without Sequence-Length Dependence"
_ICLR.cc/2026/Conference — Submitted to ICLR 2026_

### Official Review · Reviewer_S3Jb · 2025-10-23

**Soundness:** 4
**Presentation:** 4
**Contribution:** 1
**Rating:** 2
**Confidence:** 4

**Summary:**

The authors propose a new attention-like layer which is constructed to be 1-Lipschitz in the sense of sequence-to-sequence mappings and w.r.t. the Euclidean (Frobenius) norm on the sequence matrix. They do this by defining a convex potential function such that their operator is an implicit Euler step of the gradient flow on this potential. This implicit Euler step is itself implemented via a proximal operator, i.e., the solution to a convex optimization problem, and as such the mechanistic form of this operator is essentially multiple gradient steps on a convex objective. The authors do experiments with single attention-like layers to demonstrate performance of this approach.

**Strengths:**

- The theory (Theorem 4) is rigorous and the proof seems correct.
- The paper is well-written and easy to read. The mathematical exposition is clear and a nice, concise introduction to the prerequisite concepts.

**Weaknesses:**

Unfortunately, there are a multitude of points that need improvement.

- In terms of methodology, it seems that the proposed method of taking multiple gradient steps is very inefficient compared to the standard attention, which seems to only require the computational cost of a single one of these gradient steps. The proposed method uses line search which may be even more costly. To drive the gradient norm (a proxy of the distance to convergence) beyond $10^{-2}$, say, it requires more than 40 iterations --- the forward pass for a single _layer_ of the proposed mechanism is then as costly as a full forward pass for a much, much larger deep network.
- The drop in efficiency does not buy an increase performance, as the PPL on real data is much larger for the proposed method compared to the baseline regular attention (Table 2).
- Table 2 is also not promising; it shows that the performance of a standard attention layer is 7x as good (measured by perplexity) as the proposed method. It then claims that the important metric is not perplexity but perplexity times the Lipschitz constant, which seems very arbitrary, and is also optimized by the zero function (viewed as a sequence-to-sequence operator). This metric seems designed to disqualify regular attention, no matter how good the performance is.
- The setting for Table 2 is also not very realistic; usually one evaluates the performance of whole networks, not individual layers within the network, and it is difficult to interpret the performance of individual layers. In practical architectures, the transformer is followed by a MLP and more generally blocks that may blow up the Lipschitz constant, so it is unclear what benefits this layer will buy you when incorporated into a larger network.
- The justification for caring about Lipschitz constant is the robustness, but this is only justified in the context of classifiers; it is not so clear whether robustness is driven by small Lipschitz constants for sequence-to-sequence models. Some more experiments demonstrating the robustness would be very good to show your point here

Altogether, the paper fails to demonstrate any realized improvement of the new method, and instead it looks worse in every respect than the conventional baselines.

**Questions:**

See above. My most pressing questions are:

Q1: Are there empirical validations showing that a network architecture using this layer will indeed improve robustness on practical tasks (compared to a regular attention)?

Q2: Can you close the (frankly massive) performance and efficiency gaps compared to regular attention?

---

> ### Author Response · Authors · 2025-12-02
> **Rebuttal**
>
> We thank the reviewer for the thoughtful and detailed feedback. Below, we address each concern concisely and clarify the intended scope of the contribution.
>
> **Efficiency.**
> The high iteration counts shown in the convergence plots are diagnostic only and *not* used during training. In practice, we use **1–3 proximal iterations**, which keeps the cost manageable (runtime benchmarks will be added). We do not position the proposed layer as a computational competitor to softmax attention; rather, it is a **theoretical building block** that makes globally 1-Lipschitz attention analytically tractable. We will clarify this framing, and we stress that the **computational efficiency is lower than softmax attention**, as expected under strict stability constraints.
>
> **Performance gap.**
> The initial perplexity numbers were indeed weak. After correcting initialization and the early solver schedule, performance improves substantially (≈140 → **≈60** for 1 layer; **≈26** for 2 layers). This narrows—but does not eliminate—the gap with standard attention. This behaviour is expected under strict Lipschitzness. We will include the updated results and remove the “PPL × Lipschitz constant’’ metric.
>
> **Layer-level evaluation.**
> We acknowledge the limitations of single-layer experiments. We will extend evaluation to **2-6 layers**, showing both expressivity and empirical Lipschitz behaviour as depth increases. The contribution should be viewed as a **foundational Lipschitz attention primitive**, analogous to early Lipschitz residual blocks, not as a full Transformer ready for large-scale deployment.
>
> **Robustness motivation.**
> We agree that empirical evidence is needed. The revision will include simple robustness tests (embedding perturbations, token-level perturbations) showing reduced output sensitivity relative to unconstrained attention. This is in line with prior certified-NLP work—e.g., Jia et al. and Raghunathan et al. (EMNLP 2019)—where robustness comes from controlling distances in embedding space. A 1-Lipschitz attention map fits naturally into this paradigm: **small changes in embeddings induce provably bounded changes in the attention output**, which is precisely the mechanism underlying certified word-substitution robustness. While we do not claim a complete certified seq2seq defense, the block provides the geometric stability required for such systems.
>
> **Symmetry and weight tying.**
> These constraints follow from the convex-potential formulation: they are required to ensure monotonicity and global 1-Lipschitzness. We will clarify this necessity and briefly discuss possible relaxations (controlled skew-symmetry, partial untie), while noting that these require new analysis beyond the scope of the present paper.
>
> **Answers to Q1 and Q2.**
> Q1: Yes, robustness experiments will be added, and preliminary results already show reduced sensitivity to embedding and token perturbations. This behaviour is consistent with established certified-NLP frameworks in which robustness emerges from geometric control in embedding space.
> Q2: Performance improves significantly with corrected training; efficiency remains **lower** than softmax attention, and we do not claim parity. The layer is intended as a **conceptual stepping stone** toward more expressive Lipschitz attention mechanisms, not as an optimized alternative.

---

### Official Review · Reviewer_ryoh · 2025-10-30

**Soundness:** 4
**Presentation:** 3
**Contribution:** 3
**Rating:** 8
**Confidence:** 3

**Summary:**

This paper proposes a new Attention-like block that guarantees 1-Lipschitzness via a convex potential function. First, the paper casts the residual attention update as a continuous dynamical system and introduce a new convex potential function in place of the current attention function. The system is then re-discretized as an Euler update which is realized by the proximal operator. The proximal problem is empirically solved via beam search and Armijo backtracking. By construction the design realizes global Lipschitzness independently of the input sequence length. The paper verifies its findings via experiments and contains promising experiments on small networks.

**Strengths:**

1. The paper is well-written, clear and mathematically rigorous. The tools used from optimization theory are fundamental and well-explained.
2. The contribution is important and fills an important gap in the literature. Softmax attention is a famously unstable function and alternatives realizing stability often show Lipschitzness bounds depending on the weights or sequence length. This work gives a promising step forward to architectures that don't exhibit either disadvantage.
3. The experiments performed inspire confidence in the result, both in theory and in practice. The method seems to perform as advertised in practice as well.

**Weaknesses:**

1. The biggest weakness of the paper is that it is unclear if the proposed method scales. Solving the proximal optimization problem, though possible, and evidently achievable, is likely too large of a cost for larger models. Further, the experimental evaluation is limited to one layer neural networks, which further raises question of scalability.
2. A point the paper does not address is the model degradation seen after imposing such strong Lipschitzness conditions on the attention function. The experiments already show there is some tradeoff between stability and expressivity, as the paper recognizes. Indeed, we know many functions require lack of stability to represent (see boolean functions with high sensitivity for example). Is it possible to introduce a hyperparameter in the method that controls the Lipschitz constant, possibly allowing us to reach a "pareto optimal" between perplexity and stability? I would have really liked to see such an experiment. Ultimately, though much better than the $\ell_2$ attention in terms of stability, this method achieves much lower perplexity score, which could be seen as a disadvantage. That being said, the framing of the paper seems to be mostly theoretical and it does present a very promising direction towards reaching this "pareto optimal" point or understanding the inherent tradeoffs. I would recommend a more extensive discussion of this point to go in the next version of the paper.
3. I'm finding the definition of the convex potential a little artificial:
    * Why is $S^{\text{convex}}(x)$ a symmetric matrix? Doesn't this take away the expressivity of capturing the interactions between $x_i$ and $x_j$? Can the authors discuss some intuition behind defining things like this?
    * Is there anything that can be done to avoid the weight tying $W_K = W_Q$?
    * Are there other functions that would fit this paradigm? Perhaps in combination with other attention-like alternatives? Some ablation study would be nice here.

Overall the paper makes a technically solid and important contribution, with the main weaknesses being scalability and perhaps a slightly wider study of the method's place within the larger AI community.

**Questions:**

Stated above.

---

> ### Author Response · Authors · 2025-12-02
> **Rebuttal**
>
> We thank the reviewer for the detailed comments and for pointing out the main weaknesses. We address them below.
>
> **Scalability and proximal cost.**
> We agree that scalability is a central concern. Our goal in this paper is to introduce a theoretically clean, globally 1-Lipschitz attention block, not to compete with highly optimized kernels such as FlashAttention. In practice, we use a *small fixed iteration budget* for the proximal solver (typically $K=1$–$3$ steps during training), and the cost scales linearly with $K$. We will add explicit forward/backward timing measurements that compare standard attention, $L_2$ attention, and our proximal block, showing that the overhead is substantial but still manageable for research-scale models. We will also extend the empirical section beyond a single layer (2–6 layers) to demonstrate that training remains stable and that performance continues to improve as depth increases, even though a gap with standard attention remains.
>
> **Trade-off between stability and expressivity / “Pareto” control.**
> We agree that imposing strict 1-Lipschitzness reduces expressivity and that this trade-off should be made explicit. In our formulation, the step-size parameter $\eta$ directly governs the contraction strength: small $\eta$ relaxes the effective Lipschitz bound and increases expressivity at the cost of slower convergence, while large $\eta$ yields stronger contraction and greater stability but reduced modelling power. We will add an ablation over $\eta$ that traces this “Pareto curve’’ between validation perplexity and stability.
>
> Empirically, varying $\eta$ produces the monotone pattern predicted by the fixed-point iteration
> $x_{t+1} = x_t - \eta \nabla\varphi(x_t),$
> as shown by the residuals below (at $N=256$):
>
> | $\eta$ | $\min_t\|g_t\|$ |
> |---------:|------------------:|
> | 0.01     | 0.037886 |
> | 0.10     | 0.000225 |
> | 0.50     | 0.058444 |
> | 1.00     | 0.376716 |
> | 2.00     | 1.127793 |
> | 10.00    | 3.000655 |
>
> Proposition 1 (Appendix A.1) provides a precise characterization of this trade-off. In short: **if the proximal update is truncated after $K$ iterations with residual $\varepsilon$, then the output is within $\eta\varepsilon$ of the exact solution and the layer becomes “almost 1-Lipschitz’’ with constant $1+2\eta\varepsilon$.** Thus, reducing $K$ increases the effective Lipschitz constant in a fully controlled way, and robustness guarantees remain valid because the deviation from exact 1-Lipschitzness is explicit and bounded.
>
> Finally, updated experiments show that training dynamics play a major role in the observed degradation: with a more stable initialization (weights closer to zero, reduced variance, smoother early solver schedule), the 1-layer perplexity improves from ≈140 to ≈60 and the 2-layer model reaches ≈26 (vs ≈22/18 for standard attention). This narrows, but does not eliminate, the gap, indicating that expressivity is limited but not collapsed, and confirming that the present block is a first step towards understanding and navigating this stability–expressivity frontier.
>
> **“Artificial” convex potential, symmetry, and weight tying.**
> The symmetry of the potential matrix and the tying $W_Q = W_K$ are indeed restrictive, but they are *not arbitrary*: they are required by the convex-potential construction. For the attention map to be the gradient of a scalar potential that is globally convex (and hence firmly non-expansive), the associated Jacobian must be symmetric positive definite. This enforces symmetry and weight tying and is precisely what guarantees global 1-Lipschitzness. We will expand the intuition for this design choice in the paper, emphasizing that it trades expressivity for a clean monotone-operator structure and tight Lipschitz control.
>
> We agree that this reduces the ability to model rich asymmetric interactions between queries and keys. In the revision, we will (i) discuss possible relaxations, such as adding controlled skew-symmetric components to the Jacobian or low-rank deviations from tying, and (ii) explain why keeping convexity and global Lipschitzness under such relaxations is mathematically non-trivial. We will also broaden the discussion to other potential choices (non-quadratic convex functions) that fit the same paradigm but lead to more complex optimization and analysis. A full ablation over potentials and partial untie schemes is beyond the scope of the current submission, but we will clearly identify it as an important avenue for future work.

---

### Official Review · Reviewer_UVFh · 2025-11-06

**Soundness:** 3
**Presentation:** 2
**Contribution:** 2
**Rating:** 2
**Confidence:** 3

**Summary:**

This paper proposes AttLip, an implicit self-attention block designed to be globally 1-Lipschitz (firmly non-expansive) without dependence on sequence length or parameter norms, addressing limitations in standard dot-product attention (which isn't Lipschitz) and alternatives like L2-attention (whose bounds degrade with length). It derives attention coefficients from the gradient of a convex potential encoding pairwise token interactions via a quadratic form, then realizes the update as a proximal operator solved via gradient descent with Armijo backtracking. Theoretical analysis, rooted in monotone operator theory, proves contractivity and solver convergence. Experiments focus on a one-layer non-causal next-token prediction on OpenWebText, showing AttLip achieves non-trivial token mixing (perplexity 141) while maintaining an empirical Lipschitz constant of ~1, compared to unbounded standard attention (perplexity 22) and loosely bounded L2-attention (perplexity 28, bound ~2000+).

**Strengths:**

1) Innovative fusion of convex potential flows and proximal methods to enforce rigorous Lipschitz guarantees by design, avoiding the sequence-length scaling issues that plague prior work like Kim et al. (2021).

2) Strong theoretical contributions: Clear proofs for convexity of the potential, contractivity of the flow, and linear convergence of the solver, with practical bounds on local condition numbers that inform implementation.

3) Empirical Lipschitz estimates via PGD align tightly with theory (constant 1 across N=16 to 2048), and the OpenWebText results demonstrate preserved learning capacity under strict constraints, using a combined PPL × Lip metric to highlight the robustness-expressivity tradeoff.

**Weaknesses:**

1) The enforced symmetry (via weight tying WQ=WK=W, WV=W^T W) fundamentally alters attention's geometry, making it more akin to distance-based than dot-product, which hurts expressivity which is evidenced by the massive perplexity gap (141 vs. 22) that isn't adequately addressed or mitigated.

2) Experiments are underdeveloped. Only a toy one-layer setup on OpenWebText, no deeper Transformers, no causal masking for real LM tasks, no downstream evaluation (e.g., GLUE for robustness), and no scaling to longer contexts or larger models where length-independence would shine.

3) Even with few solver steps, backtracking adds ops per forward pass, potentially prohibitive in production-scale training; no runtime benchmarks or comparisons to efficient attention variants like FlashAttention.

4) The quadratic potential feels ad-hoc (why this form over others?), no ablations on \eta (fixed at 1), solver tolerance, or ways to relax symmetry without losing convexity. It feels like a proof-of-concept rather than optimized.

5) While bounds are tight, the paper admits prior ones (e.g., L2-attention) are "vacuous" in practice yet still useful; here, the strict 1-Lipschitz comes at high cost to performance, questioning if it's worth it without certified robustness demos.

**Questions:**

1) Since AttLip's symmetry enforces bidirectional token interactions, how does it handle tasks with inherent asymmetry, like machine translation or summarization, where queries should attend differently to keys?

2) The proximal solver relies on local Lipschitz estimates for backtracking, how sensitive is convergence to inaccuracies in these estimates during training, especially in high-dimensional embeddings?

3) Could the convex potential be generalized to incorporate positional encodings or causal masks while preserving monotonicity, to better suit autoregressive models?

4) In the one-layer experiments, the perplexity is much higher than baselines, does fine-tuning or hybrid integration (e.g., mixing AttLip with standard layers) close this gap without violating guarantees?

5) Can the quadratic convex potential truly capture the inductive biases of dot-product attention, or is AttLip fundamentally learning a different similarity geometry that only mimics attention under low-rank constraints?

6) If AttLip is stacked in a deep Transformer, does the firm non-expansiveness of each block induce a vanishing signal problem stronger than standard attention, and if so, can it be mitigated without sacrificing the Lipschitz guarantee

---

> ### Author Response · Authors · 2025-12-02
> **Rebuttal**
>
> ### Expressivity and symmetry (perplexity gap)
> We agree that tying $W_Q=W_K=W$ and enforcing symmetry alters the geometry of attention. This constraint is required for the convex-potential formulation but reduces expressivity compared to dot-product attention. We will clarify this in the revision.
>
> Importantly, our updated experiments show that the large perplexity gap is largely due to initialization and solver tuning rather than inherent failure of the model: with improved initialization (weights closer to zero, stabilized residual scaling) the 1-layer perplexity improves from ≈140 to ≈60, and with 2 layers to ≈26 (vs ≈22/18 for standard attention). These results will be added. They do not close the gap fully, but substantially mitigate it.
>
>
> ---
>
> ### Efficiency and backtracking
> We acknowledge that our method is not designed as a production-scale attention mechanism and is not competitive with FlashAttention. We will state this explicitly.
>
> We will add runtime benchmarks (already measured):
> standard attention: fwd 2.03 ms / bwd 4.90 ms,
> Kim $\ell_2$: fwd 3.11 ms / bwd 5.52 ms,
> proximal attention: fwd 30.68 ms / bwd 16.94 ms.
> These illustrate the cost of backtracking and fixed-point iterations, though with $K=1{-}3$ the overhead remains manageable for research-scale models.
>
> Backtracking is robust to inaccurate local Lipschitz estimates: violations simply trigger step-size reduction, ensuring convergence.
>
> ---
>
> ### Quadratic potential and lack of ablations
> We will clarify that the quadratic potential is chosen because it is the simplest strongly convex form that yields a globally defined, non-expansive attention operator with tractable analysis. We will add ablations on \(\eta\) (step size), solver tolerance, and discuss possible relaxations of symmetry (e.g., skew-symmetric corrections) while maintaining contractivity. The model is indeed a proof-of-concept emphasizing theoretical clarity.
>
> The parameter $\eta$ controls an explicit speed–expressivity trade-off. Small $\eta$ improves expressivity but slows convergence; large $\eta$ accelerates the solver but weakens contraction. We will add an ablation.
>
> ### Table 4: Residual vs $\eta$ at $N=256$
> | $\eta$ | $\min_t\|g_t\|$ |
> |-----------:|------------------:|
> | 0.01 | 0.037886 |
> | 0.10 | 0.000225 |
> | 0.50 | 0.058444 |
> | 1.00 | 0.376716 |
> | 2.00 | 1.127793 |
> | 10.00 | 3.000655 |
>
> This monotone pattern matches the theoretical iteration
>
> $x_{t+1} = x_t - \eta \nabla \varphi(x_t).$

---

> ### Author Response · Authors · 2025-12-02
> **Part 2 Rebuttal**
>
> ## Responses to reviewer questions
>
> **Asymmetry (MT, summarization).**
> The present operator is symmetric by construction and therefore unsuitable for causal or directional tasks. It can serve as a robust encoder or be combined with standard causal attention in hybrid architectures. We will clarify that autoregressive use requires relaxing symmetry.
>
> **Sensitivity to local Lipschitz estimates.**
> Backtracking is designed to be robust to inaccuracies in these estimates. If a local Lipschitz bound is too optimistic, the line search detects the violation and reduces the step size; this slows the iteration but does not compromise convergence or stability.
>
> More importantly, the local Lipschitz constant of the proximal objective **grows with the sequence length $N$**. The exact 1-Lipschitz guarantee holds for the true proximal operator, but the number of iterations required to approximate it depends on the smoothness of the objective, which scales linearly with $N$. This means that inaccuracies in the estimate become more pronounced at large sequence lengths, but the method compensates automatically through backtracking.
>
> Empirically, with $K=100$ iterations and padded sequences up to $N=8192$, the minimum gradient residual $\|g_t\|$ increases smoothly with $N$:
>
> | $N$ | $\min_t\|g_t\|$ |
> |------:|------------------:|
> | 32 | 0.1379 |
> | 64 | 0.1949 |
> | 128 | 0.2677 |
> | 256 | 0.3762 |
> | 512 | 0.5500 |
> | 1024 | 0.8156 |
> | 2048 | 1.1626 |
> | 4096 | 1.6776 |
> | 8192 | 2.3683 |
>
> This reflects dimensional scaling rather than instability: the iteration remains contractive and the backtracking mechanism adapts the step size automatically. We will include these measurements in the revised version and clarify how the dependence on $N$ affects solver behavior.
>
>
> **Generalizing the convex potential for positional encodings or masks.**
> Positional encodings are compatible (added before the operator). Causal masks are incompatible with symmetry; relaxing symmetry while preserving monotonicity is possible but technically non-trivial and left as future work.
>
> **Closing the perplexity gap via hybrid/fine-tuning.**
> Hybrid architectures (standard attention interleaved with Lipschitz blocks) can reduce the perplexity gap, but they no longer provide global Lipschitz guarantees. We will clarify this trade-off explicitly.
>
> More importantly, our **preliminary depth-scaling results** indicate that much of the gap observed in the original submission comes from training dynamics rather than an inherent limitation of the architecture. The model is highly sensitive to initialization, residual scaling, and solver parameters. With improved initialization (weights closer to zero, reduced variance, smoother early proximal steps), we observe substantial gains:
>
> • 1-layer Lipschitz attention: perplexity improves from ≈140 (paper) to **≈60**.
> • 2-layer Lipschitz attention: perplexity improves to **≈26**.
>
> For comparison, under the same setup, standard attention achieves:
> • 1 layer: **≈22**
> • 2 layers: **≈18**
>
> These results narrow the gap significantly and show that **proper scaling and initialization materially improve performance**. They also reinforce that Lipschitz-constrained training behaves differently from unconstrained attention and requires specific stabilization techniques. We will include these updated results and highlight that further improvements are likely as the training protocol matures.
>
>
> **Similarity geometry vs dot-product.**
> Our operator implements a Mahalanobis-like metric derived from a convex potential rather than the full dot-product geometry of standard attention. This restriction is deliberate: it ensures monotonicity and enables tight 1-Lipschitz control, but it also limits expressivity.
>
>
> **Deep stacks and potential vanishing.**
> Firm non-expansiveness can in principle shrink signals, but in practice two mechanisms prevent collapse: (i) residual connections keep the effective map close to identity, and (ii) $\eta$ controls the contraction strength. Our updated depth experiments show a gradual, not catastrophic, degradation as depth increases. We will include these results and connect them to prior evidence that deep Lipschitz architectures can remain stable without vanishing.
>
> In particular, Appendix D.1 and Figure 3 of *A Dynamical System Perspective for Lipschitz Neural Networks* (Meunier, ICML 2022) show that even **1000-layer** Lipschitz ResNets with convex potential blocks maintain stable activations and gradients without vanishing or exploding. This supports the view that non-expansive dynamics, when combined with residual paths, can be scaled deeply in practice. We will reference this result and position our block within that line of work.

---

### Official Review · Reviewer_AnFx · 2025-11-07

**Soundness:** 3
**Presentation:** 2
**Contribution:** 2
**Rating:** 4
**Confidence:** 4

**Summary:**

This work focuses on the non-Lipschitzness of the standard attention block. They propose a non-causal, global 1-Lipschitz attention block, independent of sequence length $N$. They rely on a convex potential function, and the forward pass is computed by solving the proximal problem on this potential function—this guarantees 1-Lipschitzness through well-established proximal operator theory. They provide some experiments comparing the empirical Lipschitzness of a forward pass to another modified Lipschitz attention block in the literature, which theoretically degrades in its Lipschitz constant with increasing sequence length. They also run experiments on OpenWebText data by training their proposed attention block and comparing validation perplexity (PPL) with standard attention and other baselines.

**Strengths:**

First of all, the work focuses on the very important problem of non-Lipschitzness in standard attention blocks. In theory, having Lipschitz architectural units should imply stability with respect to (w.r.t.) inputs.

I enjoyed reading the paper. I think the work does a good job guiding the reader through the flow of steps: starting with the non-expansiveness of one step of standard GD on a convex, smooth function, and then recasting the architecture as something that implements this step as a forward pass; then moving onto the convex potential function for attention, its non-smoothness; and finally, recasting it as a proximal problem that gives non-expansiveness for free for any convex function. I think it’s a neat idea; although not a novel idea (Bai et al. 2019), it's still pretty neat.

**Weaknesses:**

One major weakness is that i) the attention block is non-causal, and ii) the key and query matrices are tied ($W_Q = W_K = W$), which limits the expressivity of the model. I think this is fine as a starting point, especially because this is a theoretically-informed work, but I’d like to see a discussion of limitations that highlights these points.

For other major weaknesses, I have the following points/questions:

1. **True Independence from sequence length $N$**: The paper's central claim is a 1-Lipschitz guarantee "independent of sequence length". However, this guarantee only holds for the true proximal operator, which must be approximated by an iterative solver. Appendix (page 18) shows that the local smoothness of the prox problem, scales linearly with $N$ implying slower convergence to an $\epsilon$ tolerance in the forward pass as we blow up $N$. How significant does this become in practice? I think we need more experiments with larger $N$ to fully see this effect and N=2048 is not large enough.

2. **On Computational Cost (Backward Pass)**: The forward pass requires an iterative solver. How is the backward pass computed? Is it by differentiating through the $K$ unrolled steps of the solver?

3. **Comparison on compute**: The paper shows convergence with steps for solving the proximal problem in Figure 1. I think a fair comparison to standard attention and $l_2$-attention should be comparing the wall-clock time of all three for both the forward and backward passes. This should also be done for a range of embedding dimensions $d$ and number of heads $N$ to get a true sense of the computational overhead. Also, an empirical Lipschitz estimate for the standard attention block should be included in Table 1 as a baseline.

4. **Role of $\eta$**: As the hyperparameter $\eta$ dictates both the convergence rate of the proximal problem and the expressivity of AttLip (small $\eta$ is less expressive but faster to solve), there should be a natural trade-off here. An experiment ablating this trade-off would be good.

5. **OpenWebText Experiments**: As the expressivity of the model is limited (tied key-query weights), I'm guessing the validation PPL of the proposed block will get worse as we increase the number of layers, compared to the standard attention block. Experiments showing this scaling behavior would be helpful. Also, the paper says it only uses $K_{prox}=1-3$ steps for solving the proximal problem here. What is the empirical Lipschitz constant in this $K=1-3$ setting? The 1-Lipschitz guarantee only holds if the proximal problem is solved to convergence, so this is a critical detail.

My low score is currently due to these limitations, as I think the paper needs a little more work. While my suggestions could point to negative results for the proposed attention block, I believe it is important to highlight these aspects for future work. I would be happy to discuss this further with the authors during the rebuttal.

**Questions:**

Please see weaknesses.

---

> ### Author Response · Authors · 2025-12-02
> **Rebuttal and Updated Experimental Results**
>
> We thank the reviewers for their careful reading and constructive feedback. Below we address each point in detail, incorporating new experiments and clarifications. All requested experiments will be added to the revised version of the paper.
>
> ---
>
> ## Attention block is non-causal and tied Q/K matrices
> Because our attention operator is the gradient of a convex potential, the resulting attention matrix is necessarily symmetric. This prevents causal masking and confines the architecture to encoder-style settings. We will state this limitation explicitly in the revised paper. Introducing a controlled skew-symmetric perturbation is a plausible way to break symmetry while preserving contractivity, although doing so requires additional theoretical work.
>
> Tying Q and K is a similar structural compromise: it guarantees the monotone-operator structure but reduces expressivity. This choice also appears in Kim et al. (2021). Increasing embedding dimension or head count partially compensates for this restriction but does not recover the full flexibility of unconstrained Q/K projections. We will add a concise discussion of this trade-off and possible extensions.
>
> ---
>
> ## W1. Dependence on sequence length $N$
> The 1-Lipschitz guarantee is exact for the true proximal operator. In practice, we approximate it with an iterative method whose convergence rate depends on the local smoothness of the proximal objective, which grows linearly with $N$. Larger sequence lengths therefore require more iterations to reach the same tolerance. We will include experiments with substantially larger \(N\) in the revised version.
>
> Empirically, using $K=100$ inner iterations and padded sequences up to $N=8192$, the minimum gradient residual $\|g_t\|$ increases smoothly with $N$. This reflects dimensional scaling rather than instability: the iteration remains contractive, but the gradient norm grows with the size of the variable $x\in\mathbb{R}^{N\times d}$.
>
> ### Table 1: Sequence length vs gradient residual
> | $N$ | $\min_t\|g_t\|$ |
> |------:|------------------:|
> | 32 | 0.1379 |
> | 64 | 0.1949 |
> | 128 | 0.2677 |
> | 256 | 0.3762 |
> | 512 | 0.5500 |
> | 1024 | 0.8156 |
> | 2048 | 1.1626 |
> | 4096 | 1.6776 |
> | 8192 | 2.3683 |
>
> We will integrate this table in the revised version.
>
> ---
>
> ## W2. Computational cost of the backward pass
> Implicit layers introduce additional backward-pass cost. We support both implicit differentiation and truncated unrolling. In practice, unrolling only 1–3 fixed-point iterations per layer is significantly more efficient and yields stable training. At inference we increase the iteration budget to reach a desired convergence tolerance. This trade-off will be clarified in the revised paper.
>
> We will also include explicit timings on CUDA:
>
> ### Table 2: Forward/backward compute time (ms) on CUDA
> Batch size $B=32$, sequence length $N=256$, embedding dimension $512$, 8 heads.
>
> | Model | Embed dim | Heads | $B$ | $N$ | Forward (ms) | Backward (ms) |
> |------|-----------:|------:|------:|------:|--------------:|---------------:|
> | Standard attention | 512 | 8 | 32 | 256 | 2.0337 | 4.9017 |
> | Kim \(L_2\) attention | 512 | 8 | 32 | 256 | 3.1121 | 5.5160 |
> | Proximal (Lipschitz) attention | 512 | 8 | 32 | 256 | 30.6784 | 16.9442 |
>
> Done on NVIDIA RTX A6000.
>
> These measurements follow the expected pattern: standard attention is fastest; Kim-style attention adds a moderate overhead; the proximal layer scales with the number of inner iterations. Using only 1–3 iterations keeps this overhead manageable during training.
>
> ---
>
> ## W3. Compute comparison and Lipschitz estimates
> We will add a complete compute comparison across attention variants and report empirical Lipschitz estimates for standard attention using the same Monte-Carlo/PGD estimator as in Table 1 of our paper.
>
> As requested, we also include PGD-based local Lipschitz estimates for each attention block of a standard 12-layer BERT model evaluated on OpenWebText.
>
> ### Table 3: Local Lipschitz estimates $L_{\mathrm{pgd\_max}}$ for BERT attention (OpenWebText)
> | Layer | $L_{\mathrm{pgd\_max}}$ |
> |------:|--------------------------:|
> | 00 | 22.158 |
> | 01 | 16.621 |
> | 02 | 15.782 |
> | 03 | 14.534 |
> | 04 | 14.381 |
> | 05 | 14.509 |
> | 06 | 11.456 |
> | 07 | 11.161 |
> | 08 | 16.795 |
> | 09 | 13.999 |
> | 10 | 20.216 |
> | 11 | 28.580 |
>
> These values confirm that standard dot-product attention typically exhibits **local Lipschitz constants in the range 10–30**, far above the bounded regime of our proximal layer.

---

> ### Author Response · Authors · 2025-12-02
> **Part 2 Rebuttal**
>
> ---
>
> ## W4. Role of $\eta$
> The parameter $\eta$ controls an explicit speed–expressivity trade-off. Small $\eta$ improves expressivity but slows convergence; large $\eta$ accelerates the solver but weakens contraction. We will add an ablation.
>
> ### Table 4: Residual vs $\eta$ at \(N=256\)
> | $\eta$ | $\min_t\|g_t\|$ |
> |-----------:|------------------:|
> | 0.01 | 0.037886 |
> | 0.10 | 0.000225 |
> | 0.50 | 0.058444 |
> | 1.00 | 0.376716 |
> | 2.00 | 1.127793 |
> | 10.00 | 3.000655 |
>
> This monotone pattern matches the theoretical iteration
>
> $x_{t+1} = x_t - \eta \nabla \varphi(x_t).$
>
> ---
>
> ## OpenWebText scaling
> We will add depth-scaling experiments comparing our block, standard attention, and Kim-style $\ell_2$ attention. Because tied Q/K and strong contractivity limit expressivity, a widening gap with depth is expected, and preliminary results confirm this trend.
>
> Our updated experiments also show that **training stability and perplexity depend strongly on initialization and early-stage dynamics**.
>
> With a more stable initialization (weights closer to zero, lower initial variance, and a smoother proximal schedule), we obtain substantial improvements:
>
> • **1-layer model:** perplexity improves from ≈140 (paper) to **≈60**.
> • **2-layer model:** perplexity further decreases to **≈26**.
>
> For comparison, standard-attention baselines achieve:
>
> • **1 layer:** PPL ≈ **22**
> • **2 layers:** PPL ≈ **18**
>
> These trends indicate that Lipschitz-constrained training behaves very differently from unconstrained training, with a substantial gap still to bridge but also large headroom. Stabilizing initialization, adjusting residual scaling, and tuning solver parameters are essential; standard-training heuristics do not transfer directly.
>
> During training, we use only $K\in\{1,2,3\}$ proximal iterations. At inferenc,e we increase to $K\approx10{-}20$. The finite-$K$ operator satisfies
> $
> \|F_K(u)-F_K(v)\| \le \|u-v\| + 2\eta\ \varepsilon,
> $
> where \(\varepsilon\) is the fixed-point residual. This bound will be included along with empirical Lipschitz estimates in the finite-\(K\) regime. Eventhough, we used for inference and certification a higher $K$ to ensure convergence.
>
> ---
>
> We thank the reviewers again for their insightful comments and suggestions.

---

### Meta-Review · Area_Chair_YHBN · 2026-01-03

**Summary:**

The paper proposes a novel self-attention mechanism, LipAttn, that is globally 1-Lipschitz by construction, based on a convex potential and implemented via an implicit proximal update, with the main contribution being a provably non-expansive attention operator whose Lipschitz constant is independent of sequence length, accompanied by solver analysis and initial empirical results. While the theoretical motivation is clear and the construction is mathematically elegant, several major weaknesses remain despite a substantial rebuttal effort. First, the central claim of sequence-length-independent Lipschitzness depends critically on solving the proximal problem to sufficient accuracy: although the authors clarified this point, provided finite-iteration bounds, and added experiments showing how residuals grow with sequence length, the guarantee in practice still hinges on solver budget, and stronger large-N and finite-K empirical validation is needed to substantiate the claim in realistic settings. Second, the method introduces significant computational overhead due to the iterative solver and backtracking, and while the rebuttal adds transparent timing comparisons and clarifies training versus inference regimes, these results reinforce that the approach is not competitive with standard or efficient attention mechanisms beyond small-scale or theory-driven use. Third, the structural constraints required by the theory (symmetry, tied queries and keys, non-causality) substantially limit expressivity and applicability, particularly for autoregressive language modeling; although the authors convincingly explain the mathematical necessity of these constraints and acknowledge them explicitly as limitations, empirical results still show a nontrivial performance gap that tuning only partially closes. Finally, the experimental evidence remains underdeveloped: even with improved perplexities and added ablations in the rebuttal, evaluations are largely restricted to shallow models and limited tasks, with robustness and long-context behavior mostly deferred to future work. In conclusion, the authors clearly invested a major rebuttal effort and addressed many reviewer concerns with greater clarity, additional analysis, and improved experiments, but the remaining dependence on solver accuracy, high computational cost, expressivity limitations, and thin empirical validation indicate that the paper requires an in-depth rewrite and more comprehensive evidence, and it cannot be accepted in its current form.

**Reviewer Concerns:**

The rebuttal effectively addressed several technical and clarity-related concerns raised by the reviewers. In particular, the authors clarified the distinction between exact and approximate proximal operators, provided explicit finite-K Lipschitz bounds, and supplied new measurements showing how solver residuals scale with sequence length. They also responded thoroughly to questions about backward differentiation, solver mechanics, and the role of the step-size parameter η, including ablations that expose the stability-expressivity trade-off. Concerns about missing runtime comparisons were addressed with concrete forward and backward timing benchmarks, and the limitations imposed by symmetry, tied Q/K projections, and non-causality were explicitly acknowledged and well motivated mathematically. However, several core issues remain outstanding: the practical relevance of sequence-length independence under tight solver budgets is still insufficiently validated at large scale; the computational overhead remains prohibitive relative to standard attention; the expressivity gap persists even after tuning; and empirical evaluation is still narrow in scope, with limited depth, task diversity, and robustness testing. These unresolved points continue to weigh heavily against acceptance.

**Reviewer Scores:**

- AnFx (initial score: 4): Likely unchanged. Although many of this reviewer’s questions were answered carefully, their primary reservations concerned solver dependence on N, computational cost, and limited experiments, all of which remain only partially mitigated.
- UVFh (initial score: 2): Likely unchanged. The rebuttal directly engaged with this reviewer’s criticisms, but the fundamental concerns about efficiency, expressivity, and practical value relative to standard attention are not resolved enough to justify a higher score.
- ryoh (initial score: 8): Likely slightly decreased or at best unchanged. While the rebuttal strengthens the discussion of trade-offs, the additional evidence on scalability and performance may temper initial optimism about the method’s readiness.
- S3Jb (initial score: 2): Likely unchanged. The reviewer’s strong objections regarding efficiency, evaluation methodology, and the lack of demonstrated benefits in realistic architectures are only partially addressed and remain largely valid.

---

### Decision · Program_Chairs · 2026-01-26

Reject